# Environmental stress reduces shark residency to coral reefs
Michael J. Williamson [1,2,3] ✉, Emma J. Tebbs [2], David J. Curnick [1], Francesco Ferretti[4],
Aaron B. Carlisle[5], Taylor K. Chapple[6], Robert J. Schallert[7], David M. Tickler[8], Barbara A. Block[7] &
David M. P. Jacoby [1,9] ✉

Coral reef ecosystems are highly threatened and can be extremely sensitive to the effects of climate change. Multiple shark species rely on coral reefs as important habitat and, as such, play a number of significant ecological roles in these ecosystems. How environmental stress impacts routine, site-attached reef shark behavior, remains relatively unexplored. Here, we combine 8 years of acoustic tracking data (2013-2020) from grey reef sharks resident to the remote coral reefs of the Chagos Archipelago in the Central Indian Ocean, with a satellite-based index of coral reef environmental stress exposure. We show that on average across the region, increased stress on the reefs significantly reduces grey reef shark residency, promoting more diffuse space use and increasing time away from shallow forereefs. Importantly, this impact has a lagged effect for up to 16 months. This may have important physiological and conservation consequences for reef sharks, as well as broader implications for reef ecosystem functioning. As climate change is predicted to increase environmental stress on coral reef ecosystems, understanding how site-attached predators respond to stress will be crucial for forecasting the functional significance of altering predator behavior and the potential impacts on conservation for both reef sharks and coral reefs themselves.

Over the past 20 years there has been a significant decline of coral cover across the world's coral reef ecosystems due to increases in disease, tropical cyclones, and bleaching events[1,2]. Coral bleaching can cause increased mortality, reduced coral cover, loss of structural complexity, reduced biodiversity as well as altering species and community composition and ecosystem function[3–5]. Multiple shark species are reliant on coral reefs as important habitat for feeding, breeding and as social refugia[6–8]. Consequently, climate change induced changes in coral reef habitat have the potential to significantly impact the behaviour of predators associated with reef ecosystems, such as reef sharks[9,10]. Despite widespread awareness of the perilous state of global shark populations[11], including reef sharks[12,13], the link between habitat quality, changing environmental drivers, and movement ecology, as well as how these factors interact to impact population vulnerability, remains relatively unexplored[6].

Reef sharks exhibit routine use of habitats and different ecological landscapes, as they feed, develop and reproduce[14,15]. Residency, defined as 'an individual exhibiting largely uninterrupted occupancy of a limited area

for a specified period of time'[16], is one aspect of routine animal movement which facilitates crucial ecological processes, and thus is inherently linked to habitat quality, trophic interactions and population persistence[17,18]. However, many of the drivers influencing residency in reef shark species, including environmental stress, are not well understood. Changes in reef shark behaviour may have significant implications for ecological processes, such as population dynamics[19], predator-prey landscapes[20], nutrient transfer[21], dispersal[22], and management and conservation[23]. As such, understanding species responses to disturbance and the longevity or lag in these responses, especially in light of increasing anthropogenic impacts, is becoming ever more important as we face the current biodiversity crisis[24].

Grey reef sharks (*Carcharhinus amblyrhynchos*) are an Indo-Pacific distributed shark species, commonly associated with coral reefs[7,25] and currently listed as Endangered by the International Union for the Conservation of Nature (IUCN)[26]. Grey reef sharks are site-attached, central place foragers which move periodically and predictably from a core area of residency[27]. This behavioural trait make them a good model for evaluating

[1]Institute of Zoology, Zoological Society of London, London, UK. [2]Department of Geography, King's College London, London, UK. [3]Department of Genetics, Evolution and Environment, University College London, London, UK. [4]Department of Fish and Wildlife Conservation, Virginia Tech, Blacksburg, VA, USA. [5]School of Marine Science and Policy, University of Delaware, Lewes, DE, USA. [6]Hatfield Marine Science Center, Oregon State University, Newport, OR, USA. [7]Hopkins Marine Station, Stanford University, Pacific Grove, CA, USA. [8]Marine Futures Lab, School of Biological Sciences, University of Western Australia, Perth, WA, Australia. [9]Lancaster Environment Centre, Lancaster University, Lancaster, UK. ✉e-mail: michael.williamson@ioz.ac.uk; d.jacoby@lancaster.ac.uk

**Table 1 | GLMM results following model selection and model averaging for residency in grey reef sharks ($n = 122$)**

| | Estimate | Std. error | CI | z value | p value |
|---|---|---|---|---|---|
| Intercept | −2.65 | 0.25 | −3.15, −2.16 | −10.49 | <0.001 |
| Combined environmental SE index (scaled) | −0.12 | 0.01 | −0.13, −0.09 | −10.48 | <0.001 |
| Season | | | | | |
| Wet season | −0.40 | 0.02 | −0.43, −0.36 | −21.73 | <0.001 |
| Sex | | | | | |
| Male | 0.50 | 0.26 | −0.02, 1.03 | 1.87 | 0.06 |
| Year | | | | | |
| 2014 | 0.11 | 0.09 | −0.08, 0.29 | 1.14 | 0.26 |
| 2015 | 0.62 | 0.10 | 0.43, 0.81 | 6.24 | <0.001 |
| 2016 | 0.60 | 0.10 | 0.40, 0.80 | 5.82 | <0.001 |
| 2017 | 0.69 | 0.10 | 0.48, 0.89 | 6.53 | <0.001 |
| 2018 | 0.06 | 0.10 | −0.15, 0.27 | 0.53 | 0.59 |
| 2019 | 0.27 | 0.10 | 0.06, 0.48 | 2.50 | 0.01 |
| 2020 | 0.26 | 0.11 | 0.04, 0.48 | 2.30 | 0.02 |

Conditional results are presented. Estimates with unconditional standard error, 95% confidence intervals (CI), associated $p$ values are presented.

residency, and particularly amenable to acoustic telemetry, where receivers can be located around coral reefs or atolls to monitor long-term space use within core areas and movements within shallow waters[6,28]. As reef shark species are increasingly threatened with extinction[12], long-term data from widely distributed model species, particularly how shark movement is altered with changing environmental conditions on coral reefs, is valuable for informing conservation and management strategies.

Environmental stress can be defined as negative impacts on the growth and health of ecosystems resulting from changes or extremes in environmental variables[29,30]. Coral reefs are susceptible to a number of environmental stressors[31], which in turn may impact reef shark populations. However, there can be significant inter-, and intra-regional variance in how different environmental variables drive stress on these ecosystems[32,33]. For example, an environmental stress index, based on satellite remote sensing data that allows assessment of multiple abiotic environmental stressors, recently found that sea surface temperature (SST), current and wind were the primary drivers of environmental stress in the Chagos Archipelago in the Indian Ocean, however depth and SST, and Degree Heating Weeks (DHW), SST, and current, were stronger drivers of stress on coral reefs in the Red Sea and the Gilbert Islands, respectively[32]. Composite indices such as this, therefore, capture interactive variables that may increase, or reduce, environmental stress, providing an opportunity to gain a more holistic understanding of how multiple environmental stressors on coral reefs can impact reef shark movement and residency.

This study aims to investigate how reef shark residency to remote coral reefs is influenced by environmental stress to the reef itself, using an index that balances the following remotely sensed environmental variables: cloud cover, current, depth, salinity, four metrics of SST (SST, DHW, SST anomaly, SST variability), and wind. As reef shark residency is likely to be in part influenced by changes in environmental conditions[15,34], we hypothesise that as environmental stress on coral reefs increases, reef shark residency will decrease. We suggest this is a behavioural response designed to locate more appropriate habitat, enhance resource availability (e.g., prey, physical or thermal refugia), thus decreasing residency to coral reef ecosystems.

## Results
Following data preparation and filtering, 714,810 detections from 122 grey reef sharks (81 female, 41 male) from 52 receivers were used for analysis. Grey reef shark lengths ranged from 70–159 cm with mean (SD) = 117.9 cm (19.6) (Supplementary Data 1). Residency index for grey reef sharks ranged from 0.03 to 1.00 with mean (SD) = 0.34 (0.33). Environmental Stress Exposure (SE) index values, calculated at the estimated range of each

acoustic receiver, varied from 0.03–0.60 with mean (SD) = 0.22 (0.09), on a scale of 0–1 (low to high stress).

Residuals of the global model were free from heteroscedasticity and temporal autocorrelation (Supplementary Fig. 1). Following the dredge and nesting of the global model, two candidate models were found with ΔAICc values < 2 (Supplementary Table 1). Relative importance values of environmental stress exposure (SE) index, season, sex, and year were all greater than 0, indicating they are important predictors for explaining residency in reef sharks (Table 1). Total length had a relative importance of zero and was not deemed an important predictor.

Model averaging of the two candidate models indicated that environmental SE, season, and year were all significant predictors of residency in grey reef sharks in the Chagos Archipelago (Table 1). A significant negative relationship between residency and combined environmental SE index was found (estimate = −0.1, $z = −10.48$, $p < 0.001$, Fig. 1), indicating that on average across the reefs of the northern atolls of the archipelago, grey reef sharks became less resident as environmental stress on reefs increased, particularly during strong El Niño conditions, albeit with a delay in these effects during the strong El Niño event. Kernel estimates (KUD) of core (50%), 75% and 95% space use all increased almost immediately during elevated periods of stress exposure, suggesting space use became more diffuse (Fig. 1B). The variance and standard deviation of the random factors ID and station on the logit scale were 1.90 and 1.38, and 1.58 and 1.26, respectively. Marginal $R^2$ (R2m) was 0.02 and conditional $R^2$ (R2c) 0.52, suggesting high variation between stations and individuals. Results from conditional models of the random effects and their standard deviations suggest that 56% (29/52) of receivers had residency significantly different from the intercept, with some showing increased residency (Fig. 2). A similar relationship between residency and combined environmental SE index (estimate = −0.07, $z = −4.34$, $p < 0.001$) (Supplementary Table 2) was found even after data from El Niño periods were removed, suggesting persistence of this trend even without extreme climatic events known to cause high environmental stress to coral reefs. The median duration of time spent away from the forereefs were not stochastically equal between times of low and high stress; grey reef sharks were absent for significantly longer when stress was high (Brunner-Munzel; $\hat{P}^*(1235.9) = −2.8336$, $p = 0.0047$). The probability that sharks would remain away from the forereef longer during times of stress was 0.4661 (Fig. 3A).

Cross correlations of mean environmental SE index and mean residency index indicated lagged effects (Supplementary Fig. 2) with

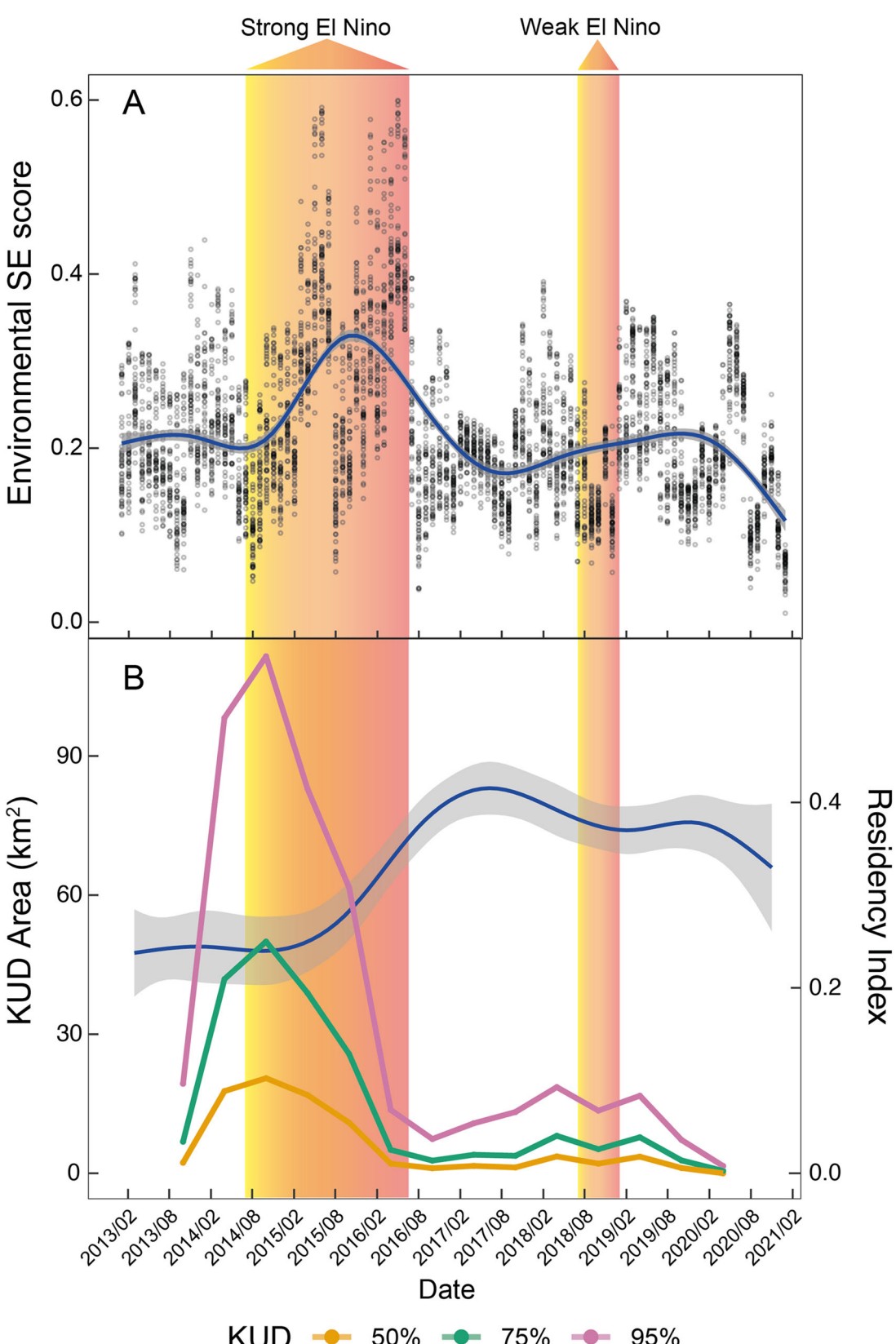

**Fig. 1 | Impact of environmental stress on shark space use and residency.** Temporal trends in combined environmental stress exposure (SE) index experienced by coral reefs in the northern atolls of the Chagos Archipelago during a strong El Niño 'episode' and weaker El Niño 'conditions' (**A**). Grey reef shark residency (blue trend line) during the same period (Feb 2013–Feb 2021) and temporal changes in area use (km²) measured as the 50% (yellow), 75% (green) and 95% (pink) kernel utilisation estimation (KUD) (**B**).

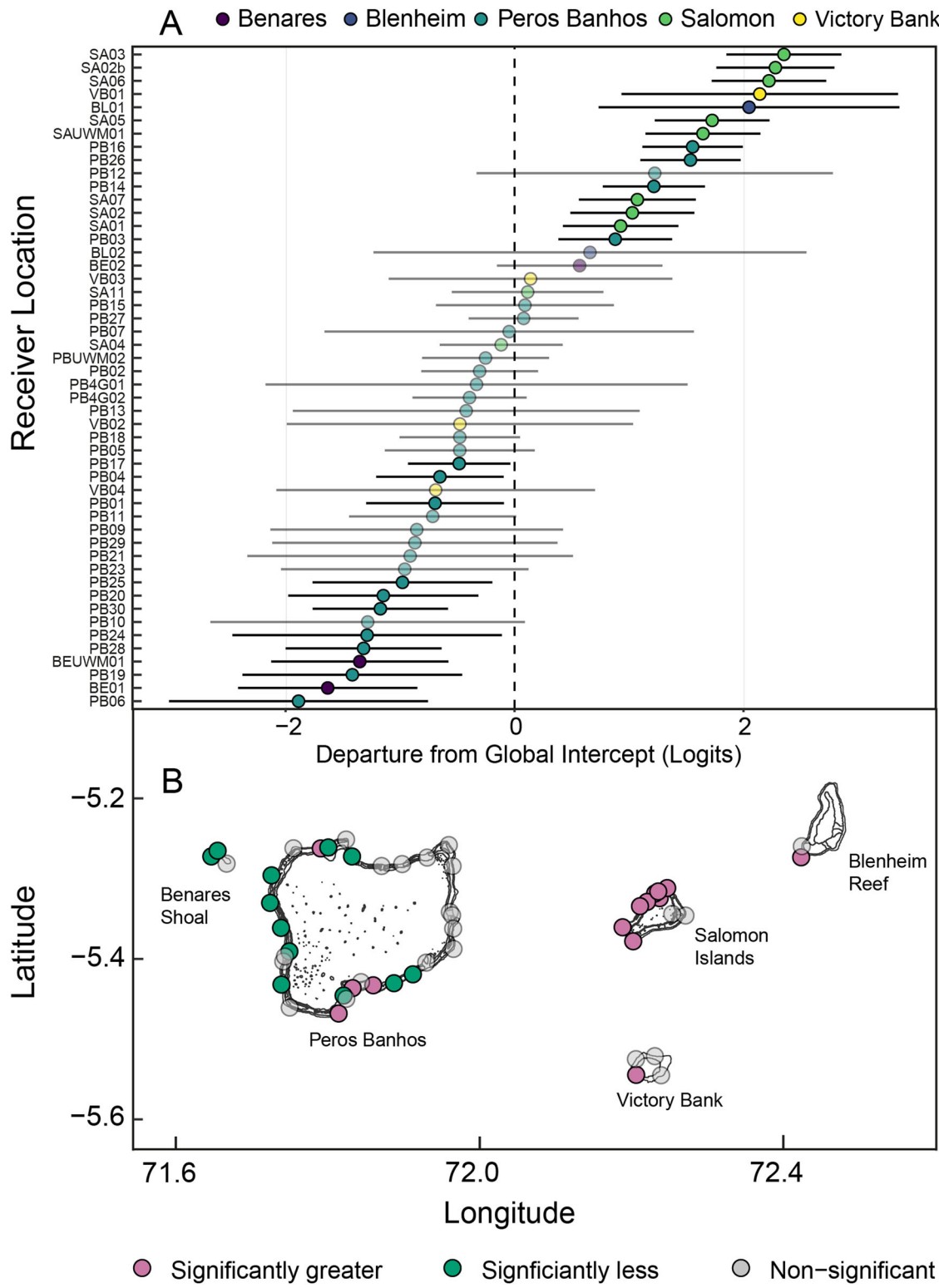

**Fig. 2 | Condition modes of random effects for each receiver location.**
**A** Departures of 122 grey reef sharks from the global intercept are plotted with 95% CIs (black bars). Receivers where CIs do not cross zero indicate average residency significantly different than the average. Receivers where grey reef sharks had less than the average residency have negative global intercept values, and those that had more have positive intercept values. **B** Spatial distribution of acoustic receivers (n = 52) coloured by departure from global intercept, where pink is significantly more resident and green significantly less resident (transparent = no change in residency).

significant negative values at $t - 0$ to $t + 16$, indicating that 'current' environmental stress on coral reefs has a significant persistent negative impact on grey reef shark residency for up to 16 months (Fig. 1). Within this time period, correlation coefficient values varied between $-0.16$ and $-0.37$. There was also a significant relationship between residency and

season, with grey reef sharks in the wet season less resident than during the dry season (Table 1) (estimate = $-0.40$, $z = -21.73$, $p < 0.001$). Residency behaviour significantly differed in all years, except 2014 and 2018, compared to the baseline year of 2013 (Table 1). Sex was not a significant predictor of residency.

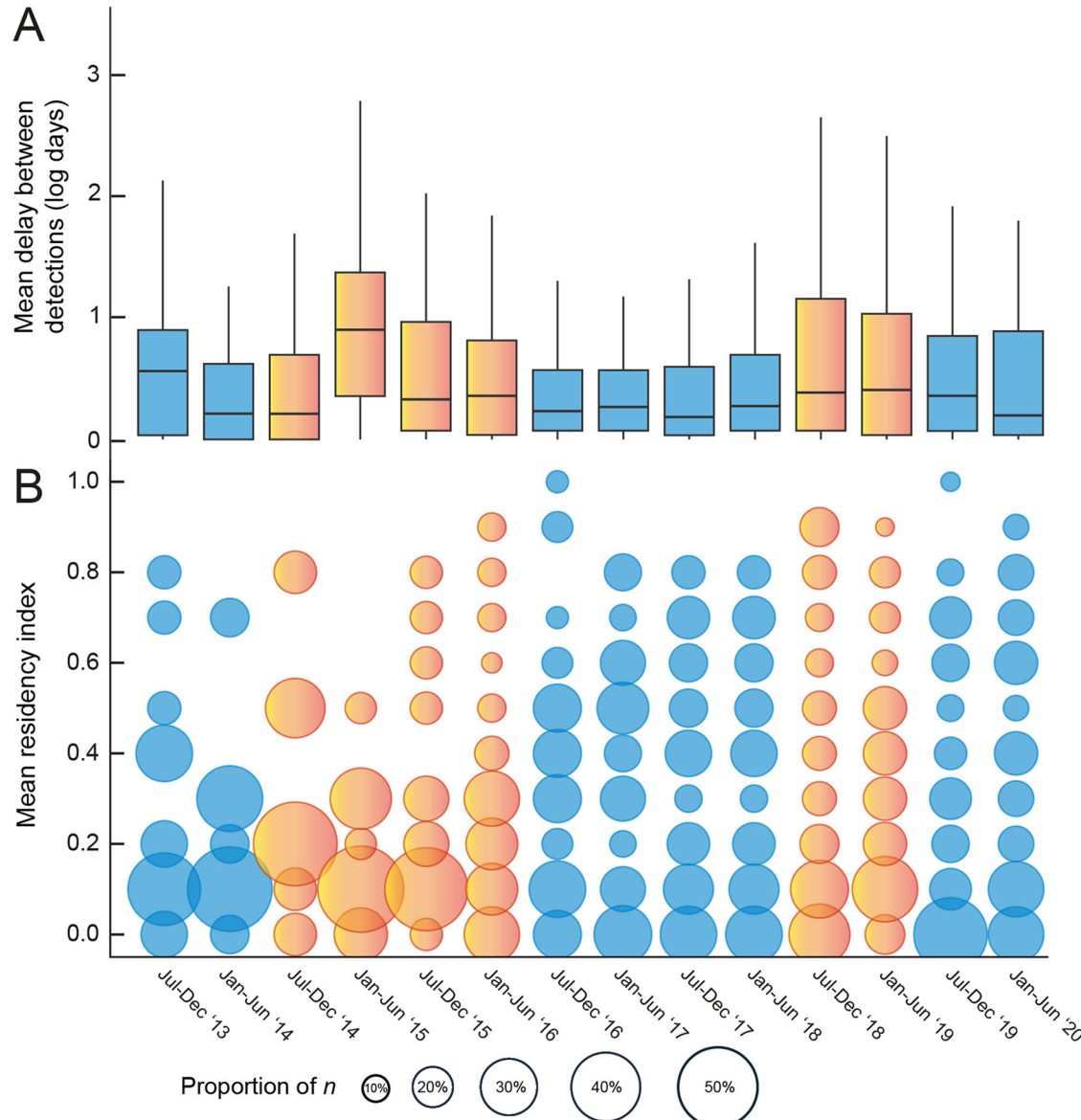

**Fig. 3 | Variation in absence and individual shark residency across an 8-year period. A** The mean delay in log days, between detections for 122 grey reef sharks leaving the forereef (note: for clarity we represent the mean but test the median using a Brunner-Munzel test to show that the probability that sharks would remain away from the forereef longer during times of stress was 0.466). Box plot represents median mean delay in log days and the interquartile range. Whiskers extend from the hinge to the highest and lowest values within 1.5× the interquartile range. Outliers are not visualised. **B** The proportion of tagged 122 grey reef sharks falling within each mean residency index bin (0.0–1.0) across 14 sixth monthly periods. Yellow/red indicate El Niño conditions and blue, non-El Niño conditions.

Generalised Additive Mixed Modelling (GAMM) results indicated a significant impact of date (Supplementary Fig. 3) (edf = 8.9, Ref.df = 9.0, $F = 209.0$, $p = <0.001$), suggesting that environmental stress varied through time in the region, lowest in March 2013, January 2017 and September 2020 and peaking in May 2015 and May 2016, matching El Niño events in the region (Fig. 1 and Supplementary Fig. 3). The adjusted R squared value was 0.28 and deviance explained 27.6%. The proportion of sharks with low residency scores also changed throughout time, with a greater proportion of sharks decreasing residency during periods of high stress also matching these El Niño events (Fig. 3B). Beta regression and post hoc results indicated significant differences between some atolls within year periods (Supplementary Table 3), with receivers at Blenheim reef experiencing significantly less environmental SE than receivers at Salomon and Peros Banhos in 2013, 2014, 2015, and 2016. Receivers at Victory Bank experienced significantly less environmental SE than receivers at Salomon in 2013, 2014, 2015, 2016 and 2018 and Peros Banhos in 2013, 2014, 2015, 2016, 2018 and 2019.

Receivers at Benares Shoal experienced significantly more environmental SE than receivers at Victory Bank in 2016.

## Discussion

Climate change is projected to have a strong influence on marine habitats and, as such, is predicted to alter and impact the movement ecology of marine species[9,10]. Here, we analysed a multi-year dataset to explore the influence of environmental habitat stress, based on a composite index of nine remotely sensed environmental variables, on the residency behaviour of a site-attached shark species, found in abundance throughout the coral reefs of the Indo-Pacific. We show that increased environmental stress on coral reef habitat reduces residency in grey reef sharks, promoting more diffuse space use and extending periods away from the reef. Additionally, our results suggest that this impact has a lagged effect across the archipelago, with increased environmental stress altering residency for up to 16 months. These findings will likely have important repercussions for trophic

interactions and reef ecosystem functioning[7,21] with potential alterations in nutrient subsidies to reefs. In addition, these results may also affect conservation and management of both grey reef sharks and coral reef ecosystems, with decreased residency potentially shifting the likelihood of interactions with both commercial and Illegal, Unregulated and Unreported (IUU) fisheries[35]. However, to date, this has yet to be examined. Interestingly, we find that these results are not ubiquitous across the whole area. Some receivers showed significant negative departures, while others showed significant positive departures, from the global mean, suggesting localised factors, such as reef resilience likely also influence residency in grey reef sharks (Fig. 2).

Our findings support our hypothesis that, overall, grey reef sharks reduce residency behaviour in the face of increased habitat stress. To respond, large mobile marine ectotherms must balance the behavioural trade-off between moving to escape stress, which requires increased energy expenditure and potentially increased risk, or remaining in the same area, which may become suboptimal but in doing so might conserve energy[36]. Although the energetic implications for these behaviours were not explicitly tested here, we do show an expansion of both core and broader space use, as well as increased periods of absence during times of stress. This might include moving into offshore, deeper and likely cooler waters, and as a result spending more time outside of receiver coverage. For a species well documented to maintain and regularly return to core areas of reef facilitating several important behavioural processes[8,27], these results are a concern. In the short term, our results suggest an immediate 'avoidance' response that has implications for this species' ecology and conservation, but in the longer term, where the benefits of departing suboptimal habitat outweigh the benefits of remaining, there are likely to be implications for the wider reef ecosystem structure (for up to 16 months post peak stress). This study does not aim to tease apart the specific mechanisms driving these short or long-term responses, which are likely different for these two processes. It does, however, offer an exciting research avenue for future studies to explore the different mechanisms influencing space use in response to environmental stress, and at different temporal scales. From a practical perspective, changes in residency may also be due to the influence of changing environmental conditions, such as wind speed, on acoustic detectability[37,38] in addition to coral reef health. Range testing was not feasible at this site during the period of study, so this could not be assessed. However, given the long time-series of data obtained, and the wide variation in environmental conditions throughout the study period, the impacts of varying detectability is like to be minimal.

Encouragingly, this negative association between stress and shark residency was not ubiquitous across all monitored locations. Model variance suggests that there is significant variation in residency at each receiver, with sharks more resident in some locations compared to others (Fig. 2). This pattern coincided with spatial and temporal variation between atolls in environmental stress exposure in this region[32,39]. Receivers in the north and west of Peros Banhos experienced a reduction in residency from the global average, but some receivers in the south of Peros Banhos and west of Salomon Islands experienced an increase in residency (Fig. 2B). Interestingly, our regional pattern of residency maps with the spatial patterns of rat infestation on the islands of the northern atolls, with receivers with higher residency overlapping with rat absent and rat eradicated islands, and receivers with significantly less residency than the global average overlapping with rat present islands[40]. Although we do not explicitly explore mechanistic drivers within this study, recent research in the Chagos Archipelago has found that seabird nutrients significantly enhance fish biomass on reefs surrounding rat absent or rat eradicated islands compared to islands with rats[41,42]. These regional patterns clearly warrant further investigation, but could indicate grey reef shark residency is also influenced by factors that make particular reef habitats more resilient to perturbation, leading to these being more stable areas to occupy as a reef predator.

The spatial variation in residency observed could also be driven by hydrodynamic factors. Coral response to environmental stress, such as bleaching, can be highly variable, even within a reef system, and often is the result of differing fine scale environmental and biological processes[32,43]. There is also some congruence between areas of increased residency and areas that are sheltered from wave exposure. Shelter from wave exposure is associated with increased coral cover and quicker recovery from bleaching events[44], another potentially important factor influencing shark behaviour. The mechanisms driving these results are clearly complex and involve a mixture of variability in shark behaviour as well as heterogeneity in coral reef response to environmental stress, and at different temporal scales.

Results from cross correlations indicate that reduced residency of grey reef sharks on coral reefs in the archipelago persists well into the future, as much as up to 16 months. Time lags in how coral reefs themselves respond to stress can be relatively short, within a few weeks[45,46], or prolonged, over periods of months or even years[47,48]. The time lag in response to stress in other reef-reliant species can also vary from months to several years[49]. For example, Halford and Caley[50] found a time lag of 12–18 months between bleaching and change in structure of fish communities on remote reefs of north-west Australia. Declines in the abundance and diversity of coral reef fishes may be apparent more than 3 years after coral depletion in some regions, due to the delay in structural collapse of dead corals[51]. The true mechanisms underlying both the instantaneous reduction in residency and its persistence for months following increased environmental stress seen here are unknown and provide an interesting next step for this research, such as exploration of possible correlation between areas of greater stress and the persistence of reduced residency in reef sharks.

This study did not examine the precise environmental factors driving reductions in residency. Stress on coral reefs is often closely linked to SST, and other temperature metrics, such as DHW and SST variability[32,52], and metrics of SST contributed considerably to the environmental SE index[32]. As such, the reduced residency found in this study could be driven by increases or changes in metrics of SST. Reef sharks are ectotherms and have been seen to exhibit behavioural thermoregulation to regulate their body temperatures and avoid physiological damage from adverse SSTs[15]. Therefore, an influence of different metrics of SST on movement is to be expected. Indeed, links with SST, SST anomalies, and SST variability and movement, residency, and presence/absence of other shark species have been seen elsewhere[53,54]. For example Ryan et al.[54] found that low SST anomalies increased white shark (*Carcharodon carcharias*) presence and residency, which increased the chances of attacks on the eastern Australian coast. However, little is known about these relationships in reef sharks[6], and the few studies that have investigated these relationships have typically found that changing environmental conditions have limited impact. For example, Schlaff et al.[55] found that size and sex were the most important drivers of activity space in Australian blacktip reef sharks *Carcharhinus melanopterus*, with salinity and water temperature having significant but relatively low impacts, while Heupel and Simpfendorfer[56] found no relationship between activity space and environmental variables in grey reef sharks on the Great Barrier Reef. As such, these results, to our knowledge, provide the some of the first evidence of changing environmental variables impacting the movement and residency of grey reef sharks.

Season was also found to have a significant effect on residency in grey reef sharks, which supports previous research at this site that showed that grey reef sharks spent more time away from reefs during the wet season compared to the dry season[28]. These changes in residency with season could be due to environmental or ecological factors. Shark species have been seen to increase movement and decrease residency during storm events[57,58], which may be increased during the wet season. Alternatively, residency changes may be due to changes in food resources, with historical fisheries known to peak in the wet season in this region[59,60]. In addition, our results confirm that year is a variable that should be regularly included as a predictor variable to account for temporal variation when modelling movement ecology of marine species[61,62], which here is most likely linked particularly to the severity of environmental change associated with El Niño events.

As climate change continues to alter oceanic conditions, environmental stress across marine ecosystems, including coral reef habitats, is predicted to increase[63,64]. Although reef sharks use coral reef systems as primary habitat, they can spend significant periods of time away from reefs for foraging[28,65], bringing substantial nutrients from deeper pelagic waters that could not be produced by the reefs themselves[21]. Consequently, reduced residency by reef sharks could lead to a reduction of subsidies between pelagic and reef ecosystems, with these cross-ecosystem flows of energy potentially influencing reef resilience during times of high environmental stress. In addition, reduced residency may have trophic implications on particular reefs, with species assemblage reorganisation possible due to loss of large mesopredators[66,67].

As well as ecological impacts to coral reef systems, these results also have implications for the ecology and conservation of reef sharks. Reduced residency may result in increased energetic costs, with subsequent impacts on survival, growth and reproduction[68-70]. In the Chagos Archipelago, and other coral reef systems, grey reef sharks are under threat from Illegal, Unregulated and Unreported (IUU) fishing activity[35,71,72], which is believed to be suppressing populations around Peros Banhos and Salomon Islands[73]. Reef shark species that are less resident on coral reef systems are more threatened with extinction risk[12]. Sharks that shift to spending more time in offshore waters may increase their vulnerability to IUU fishing, especially in a large, remote area such as the Chagos Archipelago, as increased movements in some species may increase the encounter rate with IUU vessels[35], or commercial fisheries, as they spend less time in MPAs. Alternatively, as fishers target where shark are known to aggregate[71], reduced residency may mean fewer aggregations and reduced impact of fishing on this species as they are less likely to find large numbers of them in one location.

The environmental SE index used in this study was chosen as it includes both reducers and enhancers of stress, therefore providing a balanced metric that helps decipher differences between reefs as well as providing a holistic view of multiple stressors on reef systems, not only those that drive bleaching[32]. Consequently, the results seen here may relate to reef shark response to environmental stress on coral systems, rather than to direct habitat damage. There are now remotely sensed products of coral bleaching available, such as the Allen Coral Atlas (www.allencoralatlas.org) but these were not used in this study because the data was only available for 2019 onwards. In addition, it should be noted that the environmental SE index does not contain an exhaustive list of environmental stressors. Environmental variables such as turbidity, chlorophyll-*a*, pH and ultra-violet light, all known to impact coral habitat quality, are not included in this index, due to a lack of appropriate products or due to low accuracy in shallow areas, such as coral reefs[32,74]. Furthermore, some of the variables in the environmental SE index influence habitat quality but can also have direct impacts on reef shark movement behaviour, so it is difficult to disentangle direct and indirect effects using this approach. Future work could address these issues by including additional environmental variables, currently unavailable, into the index, and when satellite-based maps of bleaching become available with sufficient temporal coverage these could be included disentangle these effects.

With climate change predicted to cause bleaching events annually by 2043[75], changing environmental stress and disturbance on coral reefs has the potential to impact the movement and ecology of reef shark species. Here, we used 714,810 detections from acoustic tagging data for 122 grey reef sharks between 2013 and 2020, combined with satellite remote sensing data to investigate how changing environmental stress on coral reefs can impact the residency of an abundant reef shark species in the Chagos Archipelago. Environmental stress, season and year were all significant predictors of grey reef shark residency. As such, we show that increased environmental stress on coral reef ecosystems reduces grey reef shark residency, expands their space use and extends periods of absence from the reef, patterns that persist more than a year beyond the main period of stress. In addition, there is also some cause for optimism with our finding that residency varies significantly at different reef locations with some areas showing increased residency. As environmental stress on coral reef systems is predicted to increase[76-78], these

changes have important consequences for the ecology and ecosystem functioning of coral reefs in the region, such as altering nutrient subsidies. Furthermore, decreased residency is likely to have conservation impacts on the sharks themselves, potentially altering their interactions with IUU and commercial fishing vessels. Finally, results from the study will enhance predictions about predator responses to climate-related stress into the future.

## Materials and methods
### Data collection and study site
The Chagos Archipelago is a large, remote archipelago, at the centre of the British Indian Ocean Territory Marine Protected Area (BIOT MPA) in the Central Indian Ocean. Established in 2010, the reefs are home to multiple resident and transient elasmobranch species[79,80]. Following El Niño events in 2015 and 2016, the coral reefs of the Chagos Archipelago experienced widespread bleaching events in response to substantial increases in environmental stress[39,79]. A long-term tagging programme of grey reef sharks has been undertaken in the region since 2013 to investigate the efficacy of the MPA for protecting large mobile fishes and for understanding how ecology can inform MPA enforcement[23,35].

Acoustic telemetry data were collected from five atolls (Benares, Blenheim, Peros Banhos, Salomon and Victory Bank) in the Chagos Archipelago between 2013 and 2021 from an acoustic array of 54 receivers[28] (Supplementary Fig. 4). All receivers were far enough apart to avoid overlap in their detection range, with mean distance to the next closest receiver being 2.15 km and ranging from 0.55–4.57 km[28]. Although range testing was not undertaken for this array, due to financial and logistical constraints of vessel time in the Chagos Archipelago, other studies conducted around coral atolls in the Indian Ocean using the same or similar equipment have found detection ranges between 300 and 500 m[81,82].

This study utilised tracking data from grey reef sharks carrying 10-year, V16, 69 kHz Innovasea coded acoustic transmitters between 1st March 2013 and the 30th November 2020. In total 122 grey reef (81 female, 41 male) were tagged with sharks being caught from handlines and barbless hooks. Larger animals (>1.5 m) were kept in the water, but all others were brought onto the boat and restrained. A seawater house was used to irrigate the gills, and a wet cloth placed over the eyes. Once restrained, sharks were measured and acoustic tags implanted intra-peritoneally through a small incision (~2–3 cm) just off the midline of their abdomen[23]. Total handling time was generally less than 5 min per animal. All procedures were approved by the Stanford University Administrative Panel on Laboratory Animal Care (APLAC) under permit APLAC-10765. Tags were configured to transmit an acoustic 'ping' containing a unique ID code with a nominal delay of 30–90 s, or 60–180 s for the duration of their battery life (~10 years), providing a long-term time-series of detection data. Differences in transmission delay between tag types were accounted for using the method by Jacoby et al.[35] to ensure that detections between the two delay types were standardised and comparable. Receivers were downloaded and serviced annually at the same time each year (March–May) with the exception of 2017, where for logistical reasons no service expedition took place.

### Data preparation
To avoid false detections from unknown tagged animals in our study system, only detections from animals with known ID codes were used for the analyses. To remove the possibility of false positives in the data set three different methods were used. First, animals with a single detection were filtered from the dataset[83,84]. Secondly, detection gaps of less than 30 s, under the ping delay of the tags, were also removed from the data by removing the second detection. Finally, transitions (movements between two different receivers) were calculated as per Williamson et al.[28] and removed if the speed of the transition exceeded 10 times the minimum sustainable swimming speeds of 0.69 m/s for grey reef sharks, resulting in a cut-off speed of 6.9 ms$^{-1}$[28,85]. To reduce any impact of the stress of capture on detected behaviour[86,87], the first 24 h of data were removed for each individual[88].

## Statistics and reproducibility

There are several equations that can be used for calculating residency from acoustic telemetry[83]. In this study, a local fixed time residency index for each shark was calculated per month at each receiver by counting the number days the shark was present per receiver in that month (minimum 2 days) divided by how many days the receiver was active during that month[83]. This allows for comparisons of residency through both time and space[83].

To explore environmental stress on the reef habitat of grey reef sharks we used the Reef Environmental Stress Exposure Toolbox (RESET) developed by Williamson et al.[32] (https://mjw1280.users.earthengine.app/view/reef-environmental-stress-exposure-toolbox). This study used nine environmental variables (cloud cover, current, depth, salinity, wind, and four SST based metrics), derived from satellite remote sensing and Google Earth engine (GEE), known to have an impact on stress and health of coral reef systems. As the spatial resolution of the nine variables varied (Supplementary Table 4), each product was resampled using bilinear interpolation to match the detection range of the receivers (500 m)[89]. These data were combined with ecological and health-based thresholds obtained from the available literature, and fuzzy logic (discontinuous functions), to develop a combined environmental SE index from satellite remote sensing data for monitoring environmental SE on coral reef systems[32]. This index was chosen, as although there are remote sensing datasets available for reef habitat and bleaching (such as the Allen Coral Atlas, https://allencoralatlas.org), these data were only available for 2019 onwards for the Chagos Archipelago. In addition, the coral reef environmental SE index incorporates several environmental variables that both enhance and reduce stress on the reef. Consequently, the index evaluates environmental stress as a whole, rather than specifically focusing on bleaching, which is one aspect of habitat stress, and can vary spatially and temporally within reef systems[48,90]. This index cannot be used to directly quantify the health of coral reefs per se. Rather, it is a temporally explicit monitoring tool (i.e., to compare against various time periods from the same region) to evaluate relative changes in stress exposure on coral reef ecosystems. The environmental SE index is scored between 0–1 with 0 being low environmental SE and 1 being high environmental SE. These index values were then included as an explanatory variable for subsequent analyses. From previous research at this site, RESET scores of 0.3 or higher indicate considerable stress to the reefs in Chagos[32].

To examine how the environmental SE index changed temporally over the study period, the environmental SE index at each receiver was included as a response variable in a GAMM, with numerical day from 01/01/2014 as an explanatory variable and 'atoll' as a random effect, using the 'gam' function in the *mgcv* package[91]. To assess how the environmental stress changed spatially, beta regression with environmental SE index as a response variable was used[92,93], with atoll as a predictor variable, using the 'betareg' function in the *betareg* package[94]. As environmental stress can change between years, year was included as an interaction in the model. Post hoc tests were undertaken between interactions using the 'emmeans' function in the *emmeans* package[95].

As all receivers were situated greater than 500 m apart[28], more than the spatial resolution of the index, detections from all receivers were included in the analysis and were not grouped. To limit exploratory analyses, and prevent model overfitting, an a priori selection of additional explanatory variables and interactions based on previous research and theory were included[96,97]. Along with the combined environmental SE index, four additional explanatory variables were included in the model. Season was included over month as season is often a more biologically relevant driver of variability in ocean systems[98]. The Chagos Archipelago has two clear seasons (wet—October to March, dry—April to September) which influences ecological processes, such as historical fisheries[59,60]. During our study period, El Niño conditions varied, and with different levels of intensity, but driving bleaching events in the region in both 2015 and 2016[39,99]. As such, year was also included as an explanatory variable, as a factor, for the global model. As both sex and size have been shown to influence movement patterns in grey reef sharks 'sex' and 'total length' were also included as explanatory variables. Continuous variables (SE index and total length) were scaled

(mean = 0, SD = 1) to aid in model fitting[97], using the 'scale' function from the base package[100]. To prevent pseudoreplication, individual ID and receiver ID were included as independent random factors in the model.

All analyses were conducted in R version 4.2.2[100]. A Generalised Linear Model (GLM) was built to assess the explanatory variables for collinearity. Collinearity was assessed by producing a variance inflation factor (VIF) from the generalised linear model using the 'check_collinearity' function in the *performance* package in R[101]. No evidence of collinearity was found, with all variables having a VIF ≤ 1.05, less than the critical threshold of 5.0[102,103]. As such, all a priori selected explanatory variables were included in the global model.

To investigate the drivers of residency, a global model, with all explanatory variables (environmental SE index, season, year, sex, and total length) and individual ID and station as random effects, was created using a Generalised Linear Mixed Model (GLMM) (family = binomial, link = logit) with the 'glmmTMB' function from the *glmmTMB* package. To fit the GLMM with proportion as the response variable, residency index was coded in the model as a fraction (days detected/days per month) and days per month used as the 'weights' argument in the model to set the basis of the response proportion[104]. Residuals of the global model were checked for heteroscedasticity, autocorrelation and data were checked for binomial distribution using the functions 'resid', 'fitted', and 'acf' from the *stats* package[100].

A model set was subsequently generated from the global model using the 'dredge' function, from the *MuMIn* package[105], with random effects as fixed terms. Models in the set were ranked by small sample size Akaike Information Criterion (AICc) values[106], and Akaike weights for each model in the confidence set were calculated[96,97,107]. To improve inference using AICc the 'nested' function from the *MuMIn* package was used on the model selection table to remove models which were more complex versions of others[96,97]. Selected models included those with ΔAICc values < 2 and that were not nested models[107]. If a single parsimonious model remained following selection, this model was fitted to the data. If no single parsimonious model subsequently resulted from the set and the weight of the best model was less than 0.9, model averaging was used[96], and the relative importance of each predictor variable calculated by summing Akaike weights for all confidence set models containing them. Model averaging was then undertaken on all predictor variables included in the final confidence set[96,97], with parameter estimates indicating the change in probability of residency as the value for continuous predictor variables increased. Categorical predictor variables were compared to the categorical variable level used as the model baseline. Positive estimates indicated an increase in residency in grey reef sharks, negative estimates a decrease in residency. It is important to note that predictors may display a high relative importance but show no significant result in the model averaged estimates, and the relative importance and model averaged estimates should be considered in combination[108]. To test whether the relationship between stress and residency were not exclusively driven by extreme stress events, a secondary analysis removing data from El Niño periods (01/07/2014–30/06/2016 and 01/07/2018–30/06/2019) was undertaken.

The effects of the fixed effects on the model, and the combination of fixed and random effects[109,110], were tested by calculating the marginal $R^2$ ($R^2$m) and conditional $R^2$ ($R^2$c) values using 'r.squaredGLMM' in the *MuMIn* package[105,110], and conditional models of the random effects, and their SDs, extracted from the top model using the 'ranef' function from the *lme4* package[111].

To if there were any lagged responses in residency behaviour as a result of occupying reefs that have undergone long-term stress (in particular during El Niño events) cross correlations were calculated from the mean monthly environmental SE index and mean monthly residency using the 'ccf' function from the *stats* package[100]. These were used to identify time lags in months ($t$) between our predictor (stress) in the present ($t = 0$) and our response (residency) into the future ($t > 0$) based on autocorrelation between the two as we move into the future.

To evaluate if grey reef sharks alter their residency through changes in space and/or time during the study, Kernel Utilisation Density (KUD)

estimates as well as the time intervals between consecutive detections were calculated. KUDs at 50, 75 and 95% were generated using the 'kernelUD' function from the *adehabitatHR* package[112] and plotted through time. The median number of days per month between detections across the array were calculated per individual. Median detection differences (i.e., detection gaps) were compared between El- Niño (elevated stress) and non-El Niño periods using the nonparametric Brunner-Munzel test[113]. Finally, residency indices were averaged across individuals for all 6 monthly periods, binned and bubble plots produced to show variation in residency across the population in response to changing stress through time.

## Data availability
Raw data supporting the results are available from the Zenodo Digital Repository: https://zenodo.org/records/11653900[114].

## Code availability
The R code used for analyses are available from the Zenodo Digital Repository: https://zenodo.org/records/11639740[115].

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

## Acknowledgements

Funding for this project was provided by the Bertarelli Foundation and contributed to the Bertarelli Programme in Marine Science. This work was also supported by the Natural Environment Research Council (Grant No. NE/L002485/1) to M.J.W., as part of the London NERC Doctoral Training Partnership at the Department of Geography, King's College London and the Institute of Zoology, London. D.J.C. was funded by Research England. All procedures were approved by the Stanford University Administrative Panel on Laboratory Animal Care (APLAC) under permit APLAC-10765. We thank the BIOT Administration for granting us permission to undertake the research. We would like thank S. Vanovac and C. Monk for their help for producing visualisations for this manuscript.

## Author contributions

Michael J. Williamson: conceptualisation (equal); formal analysis (lead); methodology (equal); visualisation (equal); writing—original draft (lead); writing—review & editing (lead). Emma J. Tebbs: supervision (equal); writing—review & editing (supporting). David J. Curnick: funding acquisition (equal); investigation (equal); supervision (supporting); writing—review & editing (supporting). Francesco Ferretti: investigation (equal); writing—review & editing (supporting). Aaron B. Carlisle: investigation (equal); writing—review & editing (supporting). Taylor K. Chapple: investigation (equal); writing—review & editing (supporting). Robert J. Schallert: investigation (equal). David M. Tickler: investigation (equal). Barbara A. Block: funding acquisition (equal); writing—review & editing (supporting). David M. P. Jacoby: conceptualisation (equal); funding acquisition (equal); investigation (equal); methodology (equal); supervision (equal); visualisation (equal); writing—review & editing (supporting).

## Competing interests
The authors declare no competing interests.
