## [peer review file · Communications Biology]

Reviewers' comments:

Reviewer #1 (Remarks to the Author):

Overall, I found this dataset detailing associations between environmental stress on coral reefs and residence of grey reef sharks, partially driven by back-to-back El Nino events during the course of the study, to be quite intriguing. I think that the results are certainly novel, in that this is the first account to my knowledge that investigates the authors' chosen environmental stress metric as a putative driver of shark residence. The data appear to be technically sound, and I think that this manuscript is important to the field of global change biology.

Major comments

I do have concerns regarding the scope of the interpretation of the studies' results. From an experimental design perspective, a weakness of the dataset is that it does not document several discrete El Nino events. The back-to-back El Nino events serve to drive variability in environmental stress, but because the dataset does not document several El Nino events during non-consecutive years, the dataset cannot demonstrate whether the association between environmental stress and shark residence is robust and repeatable. In other words, the dataset does not demonstrate that, once recovered, shark residence can be decreased again by subsequent increases in environmental stress. I understand that climate cannot be experimentally manipulated, and the shark monitoring program and environmental stress index data have not been recorded for long enough to make such claims. However, without being able to demonstrate such an association between environmental stress and shark residence, I do not think that the authors are able to provide strong evidence for their conclusions. I still think that this is an important study and a meaningful contribution to the literature, but I worry that this is an opportunistic dataset that details a one-off observation. Perhaps the authors could test whether the relationship between environmental stress and shark residence holds within years without an El Nino? This might be restricted to the first year of data, then, if the effects of back-to-back El Nino events on environmental stress and shark residence persisted for over a year. At the very least, the authors need to include a caveat addressing this matter.

Minor comments

Keywords: Consider selecting keywords that do not appear in the title.

Line 50: It might be worth mentioning that this species is listed as endangered by the IUCN.

Line 125: Grey reef sharks are distributed throughout the Indo-Pacific; they are not globally distributed.

Lines 129-132: This claim assumes that sharks will move from areas of lower exploitation to areas of higher. Is there any evidence that the reef sharks left protected habitat during the course of the study? I think it would be more prudent to claim that exploitation may change, but there are not data to indicate that exploitation would increase.

Line 138: The authors' findings 'support', but do not 'confirm', their hypothesis.

Line 196: If environmental stress is a composite metric of nine environmental variables, would it not be possible to test the influence of each of the composite parts on reef shark residency? As the authors mention, temperature is likely the predominant driver here, particularly over factors like cloud cover and wind.

Lines 213-223: Could this also be an example of temperature driving reef shark residence? I think that this is at least mentioning here, especially if reef sharks are less resident on reefs during the hotter months of the year (i.e., the wet season).

Lines 254-255: There is a word missing after 'stressed' or there is a typo.

Line 337: Consider citing Skomal et al. 2007 (Marine Technology Society Journal, 41(4): 44-48) to justify excluding the first 24 hours of data. Alternatively, cite another paper in other sharks to justify a 24-hour cutoff.

Lines 340-344: Was there a cutoff for minimum number of days sharks had to be detected for inclusion in the study? In Table S1 it looks like there were a handful of sharks that were only detected once during the entire study. Were these animals included in analyses of residence?

Figure 2 & 4: I think these could go in the supplement, they do not contribute much to the story the data tell.

Reviewer #2 (Remarks to the Author):

This was an interesting study which investigated how environmental stress may reduce reef shark residency at remote atolls. Overall, the authors have done an impressive amount of work and the implications could be wide ranging. The paper does have the potential to be of interest to a general audience. However, I feel like a greater exploration of what factors are responsible for the changes would be warranted and I wasn't entirely convinced by some of the findings.

Intro:

Line 64: I think a more detailed description of what 'environmental stress' refers to is warranted here. This is the general issue with interpretation in that a somewhat nebulous measure of stress has been applied. Not that this is in itself the wrong approach but a better understanding of what in the environmental stress influences sharks is important.

Clearly temperature or thermal anomalies, but also indirect affects such as bleaching and impacts on prey communities. However, as is, in the introduction I have no idea what environmental stress refers to.

Line 64: what does reef shark 'movement' refer to? Space use? Residency? Linear distance? Be more specific with what these predictions will be.

Results:

All shark lengths should be reported to 0.1 cm. They are not going to be accurate to 0.01cm!

Line 80 states that total length was featured in the best model but all subsequent analysis says total length was not significant?

One other issue is that I don't see the proportion variability explained by the model. I find this is really important with GAMMs as statistical significance can be obtained when very small percentages of variability is explained (and therefore not likely biologically significant).

More details about what specifically has changed regarding the sharks behavior and why are needed. For example looking at figure one it appears that what changes is the area used by the animals, rather than simple residency. The el nino years show a larger KUD. So do the sharks use a larger area during 'stressful' times or do they actually alter their residency to the atoll? This distinction needs to be clarified.

I would also be a bit concerned that some of the results are related to correlations between years. For example, Jan-Jun 2014 was not an El Nino year, yet the UD sizes and residency analysis (Fig 1 and Fig 3) are almost identical to the El Nino year results. This may suggest that 2014-2016 results may have been driven by some other factor. Was 2013 an El Nino year?

Is Fig.4 needed? I am not sure what additional information is gained

Discussion:

Line 203: I would say there is a lot of individual variability with depth use but some grey reef sharks may spend considerable time below 50 m (see Papastamatiou et al. 2018).

The final big issue is it would be nice to try and see what specifically from the 'stress' measure is causing changes in shark behavior. While I understand why there is the need for a single metric, it becomes less informative when the specific variables playing a role are unknown. As an example, cloud cover (one of the variables included) is not likely to cause changes in shark residency. While they do say they cant tease those apart, do they have access to he raw data that went into the single metric? Or is the index the only processed value they received.

Reviewer #3 (Remarks to the Author):

This manuscript uses acoustic telemetry data from grey reef sharks tagged in the Chagos Archipelago and satellite derived environmental data that is indicative of coral reef stress to test the relationship between the residency of sharks and the stress experienced by coral reefs. Given the increasing stress that climate change is placing on coral reefs, understanding such effects will be of importance to future management of sharks in reef environments. I consider the premise of the manuscript to be a good one – this is a sensible concept to be investigated. However, I think that there is a little more work to do to get the manuscript to a point where it makes a compelling case for the relationship that was identified. My major points are:

A. The SE Index (environmental stress indicator) is a set of environmental parameters that appear to be used to show the level of stress that a coral reef is facing. As used in the current manuscript the index is calculated at the smallest possible spatial scale (50 m radius) to correspond to the location of acoustic receivers (more on the spatial scale of this coverage later). Where I am struggling is that nowhere that I could find in the manuscript was there any calibration as to what a particular level of the index means. For example, is there a value at which bleaching would occur? Ultimately, the reader has no way of knowing what a value of 0.2, or 0.4, of the index means to a reef. One is obviously worse than the other, but is there a linear relationship in this relationship? The addition of this type of information is critical to the reader understanding what the index means and so they would be able to get a sense of what this might mean for the sharks. From fig 1 it appears that the mean(?) Values do not exceed 0.4, and are often around 0.2. what I really wanted to know was how bad was 0.4 for reefs?

B. As mentioned above, the pixel size for the index was 50 m to represent the location of each of the receivers. However, the manuscript talks about the likely acoustic detection range being 300-500 m (radius). I wonder whether calculating the index on the area that corresponds to the detection range would be more appropriate since this is the area that sharks detected would occur in. I suspect this will make little difference since the index is likely to vary little over such small spatial scales. However, I could be wrong, and I think it would be useful to explore using this slightly broader area for the index.

C. I'm struggling with the kernels shown in fig 1. I assume that these are the kernel distributions of the shark locations of those detected in the relevant time period. This is a pretty standard approach. However, the figure caption says that these show "shark residency behavior". However, I don't see the direct connection here. Assuming these are occurrence data it shows where the animals were mostly detected, not what their residency was. Given the detection ranges of the receivers the kernels overestimate the

space use of the sharks, especially when sample sizes are small (up until July-Dec 2015). By this I mean that the areas shown in the kernels include mostly area that is not covered by receiver detection ranges. The change in size of the kernels with sample size could be overcome by using a fixed smoothing factor (h) for the kernel estimation (ie the same for all periods rather than using an estimator that changes with each time period). The issue that I see here is that some readers may have the impression that the sharks are more widely moving in the early period dominated by El Nino conditions because of the larger size of the kernels. Given the sample size and smoothing factor issues this is not a certainty. However, I think the bigger question is what it is the kernels are meant to be showing and whether they have a place in the manuscript. It remains unclear to me how they help the reader interpret the residency index values. Thus at the very least there needs to be more information provided on what they show, and what it means for the results on residency that they are meant to demonstrate.

D. Related to the sample size issue identified above (ie smaller sample sizes up to Jul-Dec 2015) is what is displayed in fig 3. There is a clear change in distribution and size of the points when sample sizes increase. Given this I think it is hard to infer too much from this figure and I suggest it would be better to plot this data in another way that is not affected by sample size, and that would allow for some statistical testing of the distribution of the values between periods. It is possible that there is a distinct change in residency index distribution during this period, but without removing the sample size effects it is hard to conclude such a result.

E. One of the important considerations in this work is where do the sharks go when they are not detected by the receivers? This is explored a little in the discussion, but I think is a more central part of the consideration of what this all means. And it may also be an area that further analysis might be able to help reveal. The manuscript implies that sharks travel considerable distances when environmental stress increases. This is not an unreasonable conclusion, but not one that is well explained or well tested in the manuscript. The difficulty is that the acoustic receivers only provide particle presence data and there is no information on where they have gone. So it is equally plausible that when a shark is not detected that it is just outside the detection range of the receiver (e.g. moved down the reef into deeper water where the receiver cannot hear the signal), within the detection range but not detected for some reason (e.g. signal interference, environmental conditions) or a long way from the receiver. Thus the conclusion that sharks are traveling considerable distances is one of a number of equally plausible conclusions. I see a couple of options for how to dig more deeply into this issue. Firstly, the residency index is calculated individual receivers, and measures the proportion of days each month that an individual is present. So it would also be possible to calculate the mean/max/etc number of continuous days absent by month. If there is an increase in the value of this index, then I would interpret this as the

likelihood that sharks have moved farther away from the receiver. However, if this value does not change, then it suggests that sharks are moving away for similar periods and possibly similar distances as when they are more resident. The other potential test would be to determine the likelihood that individuals moved to other receivers during periods of lower residency. If the animals are detected at other receivers and farther away during periods of lower residency this would be good evidence of movement of considerable distances. By considering both of these analyses in combination, you could get an indication of where they may have moved (a) if they were detected at other receivers this might suggest that they are moving to other reef locations (possibly with lower env. stress), (b) if they did not get detected at other receivers, but did have longer periods of continuous absence this might indicate movement to areas farther from the reef or into deeper water. The depth question might be answerable if the data had depth sensors, but the methods do not mention them so I assume they did not and so this could not be investigated. This type of exploration would provide a lot more depth to this manuscript and help turn the Discussion from what is largely speculative about a range of issues that are currently untested.

F. Related to the previous point is the possibility that the residency of sharks did not change over time, but their detectability did (and hence the number of days detected per month). The variation in the detection ability of acoustic receivers with a wide range of environmental conditions has been tested a reasonable amount (e.g. Huveneers et al., 2016) and shown to vary quite a lot in certain situations. As such this is a factor that should be addressed in the current manuscript. For example, there was a season difference in residency according to the model. I assume one of these was a windy monsoon season. Wind driven wave action can be a factor that effects detectability, and so may have been a factor. Of course it could also be the opposite, so context here is important. Testing these issues can be quite tricky, and so I am not suggesting to do much beyond acknowledging that this may be a factor that needs to be considered in the interpretation of the data. The authors may have some way to test this as well, which would be even better.

G. As mentioned above, the Discussion is largely speculative because of the types of data and results generated. Hence the suggestion that some additional exploration of the data be undertaken to help provide some way of reducing this speculation. The temptation is always to assume that there is a important an meaningful reason for the patterns found, but it is important to try and provide some additional information that may support these over other types of conclusions. I'm not suggesting that all speculation is removed form the Discussion. Some of this is useful and warranted. However, it needs to be presented in a way that the reader understands that there a range of plausible outcomes. So statements such as "These findings will likely have important repercussions for trophic interactions and reef ecosystem functioning" (line 128) should be tempered a bit and the full range of

possibilities provided.

H. In the paragraph starting on line 196 the effect of environmental factors on the movement and residency of sharks is explored. This is a useful paragraph. However most of the examples come from non-reef species. The statement “little is known of these relationships in reef sharks” is made. While there is not extensive knowledge there are some studies and I think these should be considered here. Some of these conclude that environmental parameters have limited effects on reef sharks, which would seem like a useful piece of information for the reader to have. Examples that I am familiar with include: (Heupel & Simpfendorfer, 2015), (Schlaff et al., 2017), (Espinoza et al., 2015). On line 217 the observation is made that sharks increase movement during storm events. This is mostly true, but Udyawer et al. (2013) reported that a reef sharks species (blacktip reef sharks) did not move during a major storm when all other species did.

I believe that addressing the above points will significantly improve the manuscript and provide more certainty about what the results show and how important they are for coral reef sharks.

Espinoza, M., Heupel, M. R., Tobin, A. J., & Simpfendorfer, C. A. (2015). Residency patterns and movements of grey reef sharks (*Carcharhinus amblyrhynchos*) in semi-isolated coral reef habitats [Article]. *Marine Biology*, 162(2), 343-358.

Heupel, M. R., & Simpfendorfer, C. A. (2015). Long-term movement patterns of a coral reef predator. *Coral Reefs*, 34, 679-691. <https://doi.org/10.1007/s00338-015-1272-4>

Huveneers, C., Simpfendorfer, C. A., Kim, S., Semmens, J. M., Hobday, A. J., Pederson, H., Stieglitz, T., Vallee, R., Webber, D., Heupel, M. R., Peddemors, V., & Harcourt, R. G. (2016). The influence of environmental parameters on the performance and detection range of acoustic receivers. *Methods in Ecology and Evolution*, 7(7), 825-835.
<https://doi.org/10.1111/2041-210X.12520>

Schlaff, A. M., Heupel, M. R., Udyawer, V., & Simpfendorfer, C. A. (2017). Biological and environmental effects on activity space of a common reef shark on an inshore reef. *Marine Ecology Progress Series*, 571, 169-181. <http://www.int-res.com/abstracts/meps/v571/p169-181/>

Udyawer, V., Chin, A., Knip, D. M., Simpfendorfer, C. A., & Heupel, M. R. (2013). Variable

response of coastal sharks to severe tropical storms: environmental cues and changes in space use [Article]. *Marine Ecology Progress Series*, 480, 171-183.

COMMSBIO-23-4858-T Reviewer Comments

Reviewer #1 (Remarks to the Author):

Overall, I found this dataset detailing associations between environmental stress on coral reefs and residence of grey reef sharks, partially driven by back-to-back El Nino events during the course of the study, to be quite intriguing. I think that the results are certainly novel, in that this is the first account to my knowledge that investigates the authors' chosen environmental stress metric as a putative driver of shark residence. The data appear to be technically sound, and I think that this manuscript is important to the field of global change biology.

We thank the referee for their insightful comments throughout and for this positive feedback. We have attempted to address all of the concerns and have undertaken substantial revisions delving even deeper into these interesting results. The feedback from this and the other two referees has really helped to improve the manuscript.

Major comments

I do have concerns regarding the scope of the interpretation of the studies' results. From an experimental design perspective, a weakness of the dataset is that it does not document several discrete El Nino events. The back-to-back El Nino events serve to drive variability in environmental stress, but because the dataset does not document several El Nino events during non-consecutive years, the dataset cannot demonstrate whether the association between environmental stress and shark residence is robust and repeatable. In other words, the dataset does not demonstrate that, once recovered, shark residence can be decreased again by subsequent increases in environmental stress. I understand that climate cannot be experimentally manipulated, and the shark monitoring program and environmental stress index data have not been recorded for long enough to make such claims. However, without being able to demonstrate such an association between environmental stress and shark residence, I do not think that the authors are able to provide strong evidence for their conclusions. I still think that this is an important study and a meaningful contribution to the literature, but I worry that this is an opportunistic dataset that details a one-off observation. Perhaps the authors could test whether the relationship between environmental stress and shark residence holds within years without an El Nino? This might be restricted to the first year of data, then, if the effects of back-to-back El Nino events on environmental stress and shark residence persisted for over a year. At the very least, the authors need to include a caveat addressing this matter.

We thank the reviewer for their point. We agree that it is important to tease out whether this is a consistent relationship or whether it is driven by extreme events, such as El Niño events. We have now added additional years of data to the study to address this concern. The data now run from April 2013 to September 2020 and the number of detections has increased from 417,158 to 714,810. This incorporates more periods of 'non-El Niño' conditions. In rerunning our models, we found our model results to be similar, albeit with a slightly different model estimates (-0.12 compared to -0.1), R2m (0.020 compared to 0.017) and R2c (0.524 compared to 0.519), which increase the strength of this association. To improve interpretation, we also ran a dataset that was filtered to remove data from El Niño periods (07/2014 – 06/2016 and 07/2018 – 06/2020). The relationship between stress and residency persisted, even when data from El Nino periods were excluded, with increased environmental stress decreasing residency in grey reef sharks. As such, we feel this provides additional evidence that this relationship isn't purely driven by the extreme El Nino events.

We have added this extra analysis to our methods (line 482) results (lines 116) and added the model to tables to supplementary material (Supplementary Table 3).

Line 482 – To test whether the relationship between stress and residency were not exclusively driven by extreme stress events, a secondary analysis removing data from El Niño periods was undertaken.

Line 116 - A similar relationship between residency and combined environmental SE index (estimate = -0.07, z = -4.34, p <0.001) was found even after data from El Niño periods were removed,

suggesting persistence of this trend even without extreme climatic events known to cause high environmental stress to coral reefs.

Minor comments

Keywords: Consider selecting keywords that do not appear in the title.

Good suggestion, thanks. We have now replaced keywords that overlap with the title (which has also changed slightly).

Line 50: It might be worth mentioning that this species is listed as endangered by the IUCN.
This has been added as suggested.

Line 125: Grey reef sharks are distributed throughout the Indo-Pacific; they are not globally distributed.

Thanks for spotting this! It was a mistake on our part which has now been corrected.

Lines 129-132: This claim assumes that sharks will move from areas of lower exploitation to areas of higher. Is there any evidence that the reef sharks left protected habitat during the course of the study? I think it would be more prudent to claim that exploitation may change, but there are not data to indicate that exploitation would increase.

We thank the reviewer for their comment. We do not have any evidence that any animals left the MPA during the course of the study, but we have now added an analysis that quantifies gaps in the acoustic data (i.e. how long individuals go undetected for when they appear to leave the reef-based receiver array). We have now modified this sentence to reduce subjectivity, see line 166.

Line 166 - In addition, these results may also affect conservation and management of both grey reef sharks and coral reef ecosystems, with decreased residency potentially shifting the likelihood of interactions with both commercial and Illegal, Unregulated and Unreported (IUU) fisheries³⁵. However, to date, this has yet to be examined.

Line 138: The authors' findings 'support', but do not 'confirm', their hypothesis.
This has now been corrected.

Line 196: If environmental stress is a composite metric of nine environmental variables, would it not be possible to test the influence of each of the composite parts on reef shark residency? As the authors mention, temperature is likely the predominant driver here, particularly over factors like cloud cover and wind.

This is a good question and one that we have spent a long time thinking about. There is significant variance both within and between the variables that drive environmental stress on coral reefs. Our previous work (Williamson et al. 2022) showed that the environmental variables that drive stress vary at different sites (see Figure RR1 below). On the reefs of the Chagos Archipelago, although SST was an important driver, wind, cloud cover, current and depth, also contributed significantly to environmental stress (see Figure RR2 below). The ways in which different environmental 'enhancers' and 'reducers' of stress combine to impact coral reefs are complex and variable even on a local scale (see Figure RR2, as well as Head et al. 2019 who document variable and taxon-specific responses of the reefs in Chagos to stress, even within atolls). The use of an index allows us to standardise our understanding of stress across the areas of interest, as reefs within and between regions may have different responses to SST variability for example.

We do however, feel this detail is lacking in the manuscript and as such have added a paragraph to the introduction outlining these points and discussing this need for a composite metric. See line 63.

Line 63 - Environmental stress can be defined as negative impacts on the growth and health of ecosystems resulting from changes or extremes in environmental variables^{29,30}. Coral reefs are susceptible to a number of environmental stressors³¹, which in turn may impact reef shark populations. However, there can be significant inter-, and intra-regional variance in how different

environmental variables drive stress on these ecosystems^{32,33}. For example, an environmental stress index, based on satellite remote sensing data that allows assessment of multiple abiotic environmental stressors, recently found that sea surface temperature (SST), current and wind were the primary drivers of environmental stress in the Chagos Archipelago in the Indian Ocean, however depth and SST, and Degree Heating Weeks, SST, and current, were stronger drivers of stress on coral reefs in the Red Sea and the Gilbert Islands, respectively³². Composite indices such as this, therefore, capture interactive variables that may increase, or reduce, environmental stress, providing an opportunity to gain a more holistic understanding of how multiple environmental stressors on coral reefs can impact reef shark movement and residency.

Figure RR1. Biplots of the different samples of reefs and regions from PCA of SE scores for environmental variables used in the RESET. X and Y axes show normalized principal component scores (taken from Williamson et al. 2022).

Figure RR2. Box plots of the distribution of variables across SE scores between different sites in the central Red Sea, the Chagos Archipelago and the Gilbert Islands (taken from Williamson et al. 2022).

Williamson MJ, Tebbs EJ, Dawson TP, Thompson HJ, Head CEI, Jacoby DMP. 2022 Monitoring shallow coral reef exposure to environmental stressors using satellite earth observation: the reef environmental stress exposure toolbox (RESET). *Remote Sens. Ecol. Conserv.* 8:855-874. (doi:10.1002/rse2.286)

Head CEI, Bayley DTI, Rowlands G, Roche RC, Tickler DM, Rogers AD, Koldewey H, Turner JR, Andradi-Brown DA. 2019 Coral bleaching impacts from back-to-back 2015–2016 thermal anomalies in the remote central Indian Ocean. *Coral Reefs* 38, 605–618. (doi:10.1007/s00338-019-01821-9)

Lines 213-223: Could this also be an example of temperature driving reef shark residence? I think that this is at least mentioning here, especially if reef sharks are less resident on reefs during the hotter months of the year (i.e., the wet season).

This is a good point. Temperature can be an important driver of reef shark ecology. As such we have added information in this section to discuss this point, line 245.

Line 245 - As such, the reduced residency found in this study could be driven by increases or changes in metrics of SST. Reef sharks are ectotherms and have been seen to exhibit behavioral thermoregulation to regulate their body temperatures and avoid physiological damage from adverse SSTs¹⁵. Therefore, an influence of different metrics of SST on movement is to be expected. Indeed, links with SST, SST anomalies, and SST variability and movement, residency, and presence/absence of other shark species have been seen elsewhere^{53,54}. For example Ryan et al.⁵⁴ found that low SST anomalies increased white shark (*Carcharodon carcharias*) presence and residency, which increased the chances of attacks on the eastern Australian coast. However, little is known about these relationships in reef sharks⁶.

Lines 254-255: There is a word missing after 'stressed' or there is a typo.
This has been corrected to 'stress'. Thanks.

Line 337: Consider citing Skomal et al. 2007 (Marine Technology Society Journal, 41(4): 44-48) to justify excluding the first 24 hours of data. Alternatively, cite another paper in other sharks to justify a 24-hour cutoff.
Added as suggested.

Lines 340-344: Was there a cutoff for minimum number of days sharks had to be detected for inclusion in the study? In Table S1 it looks like there were a handful of sharks that were only detected once during the entire study. Were these animals included in analyses of residency?
Animals with single detections were removed from the dataset, one of three methods used to remove false positives from the dataset. This detail can be found on line 376. An animal had to be detected two days in that month at a single receiver for residency to be calculated for that individual. This information has now been elaborated on in the manuscript (line 387).

Line 376 - First, animals with a single detection were filtered from the dataset^{81,82}.

Line 387 - In this study, a local fixed time residency index for each shark was calculated per month at each receiver by counting the number days the shark was present per receiver in that month (minimum two days) divided by how many days the receiver was active during that month⁸¹.

Figure 2 & 4: I think these could go in the supplement, they do not contribute much to the story the data tell.

We believe Figure 2 gives a useful spatial representation of the data, showing that this overall effect of stress on residency varies at different parts of the reef / atoll. This lays the groundwork for more focused studies that explore what it is about those areas that might be buffering the impacts of stress to sharks (or the particular section of reef they associate with). As grey reef sharks are central place foragers and consequently show high site fidelity, this next step linking to particular habitat characteristics will be important (but goes beyond the scope of the current manuscript). However, we agree with the reviewer in regards to figure 4 and have added this to supplementary materials.

Reviewer #2 (Remarks to the Author):

This was an interesting study which investigated how environmental stress may reduce reef shark residency at remote atolls. Overall, the authors have done an impressive amount of work and the implications could be wide ranging. The paper does have the potential to be of interest to a general audience. However, I feel like a greater exploration of what factors are responsible for the changes would be warranted and I wasn't entirely convinced by some of the findings.

We thank the referee for this insightful feedback which has really helped strengthen the revision of this work. Through substantial revision, we believe that we now offer much more both in terms of the stress index but also in trying to explain what the sharks are actually doing in response to stress. We also include an additional two years of data to help strengthen our claims.

Intro:

Line 64: I think a more detailed description of what 'environmental stress' refers to is warranted here. This is the general issue with interpretation in that a somewhat nebulous measure of stress has been applied. Not that this is in itself the wrong approach but a better understanding of what in the environmental stress influences sharks is important. Clearly temperature or thermal anomalies, but also indirect affects such as bleaching and impacts on prey communities. However, as is, in the introduction I have no idea what environmental stress refers to.

We thank the reviewer for this point. In response to comments from multiple referees, we have added information into the introduction defining our environmental stress index more clearly and what it

means for coral reef systems (line 63). We include more of a rationale for using an index rather than individual variables and also add more detail in the discussion around the individual environmental stressors that could be driving this result. But yes, we agree, temperature is clearly playing a big part (see also our response to the previous referee comments on this).

Line 63 - Environmental stress can be defined as negative impacts on the growth and health of ecosystems resulting from changes or extremes in environmental variables^{29,30}. Coral reefs are susceptible to a number of environmental stressors³¹, which in turn may impact reef shark populations. However, there can be significant inter-, and intra-regional variance in how different environmental variables drive stress on these ecosystems^{32,33}. For example, an environmental stress index, based on satellite remote sensing data that allows assessment of multiple abiotic environmental stressors, recently found that sea surface temperature (SST), current and wind were the primary drivers of environmental stress in the Chagos Archipelago in the Indian Ocean, however depth and SST, and Degree Heating Weeks, SST, and current, were stronger drivers of stress on coral reefs in the Red Sea and the Gilbert Islands, respectively³². Composite indices such as this, therefore, capture interactive variables that may increase, or reduce, environmental stress, providing an opportunity to gain a more holistic understanding of how multiple environmental stressors on coral reefs can impact reef shark movement and residency.

Line 64: what does reef shark 'movement' refer to? Space use? Residency? Linear distance? Be more specific with what these predictions will be.

This is a valid point. In this new iteration of the manuscript we now make the distinction between residency (using residency indices) and space use (measure through our KUD analyses). We recognise that it is important to make it clear that residency can still reduce even if sharks are remaining within the array (particularly in site affiliated species like grey reef sharks). Importantly we wanted to be able to test responses of both these elements of movement in relation to stress so have included additional analyses to this effect (summarised in a new Fig. 1). The manuscript has been modified throughout to reflect these changes.

Figure 1. Impact of environmental stress on shark space use and residency. A Temporal trends in combined environmental stress exposure (SE) index experienced by coral reefs in the northern atolls of the Chagos Archipelago during a strong El Niño ‘episode’ and weaker El Niño ‘conditions’. B Grey reef shark residency during the same period (Feb 2013 – Feb 2021) plotted against changes in area use (km²) measured as the 50% (yellow), 75% (green) and 95% (pink) kernel utilization estimation (KUD).

Results:

All shark lengths should be reported to 0.1 cm. They are not going to be accurate to 0.01cm!

We thank the reviewer for this point and have altered this in the manuscript, see line 90.

Line 90 - Grey reef shark lengths ranged from 70 – 159 cm with mean (SD) = 117.9 cm (19.6) (Supplementary Table 1).

Line 80 states that total length was featured in the best model but all subsequent analysis says total length was not significant?

Yes, this is correct. In this manuscript a model averaging approach was taken to find a selection of models that best fit the data rather than selecting the ‘top’ model with the lowest AIC. As such, predictor variables, can be deemed as important to the system, without being significant. We note that we did not include an explanation of this in our methods. We have added further detail to the methods to clarify this, see line 480.

Line 480 - It is important to note that predictors may display a high relative importance but show no significant result in the model averaged estimates, and the relative importance and model averaged estimates should be considered in combination¹⁰⁶.

One other issue is that I don't see the proportion variability explained by the model. I find this is really important with GAMMs as statistical significance can be obtained when very small percentages of variability is explained (and therefore not likely biologically significant).

This is an important point. We used generalised linear mixed models rather than GAMMs for our residency model. To assess variance and impact driven by fixed and random effects we calculate marginal and conditional values, which are included in the manuscript (line 112). We do use a GAMM to assess the change in environmental stress exposure through time at the site. We have now included the deviance explained and the adjusted R squared value in our results for this particular analysis (line 144).

Line 112 - Marginal R² (R²_m) was 0.02 and conditional R² (R²_c) 0.52, suggest high variation between stations and individuals.

Line 144 - The adjusted R squared value was 0.28 and deviance explained 27.6%.

More details about what specifically has changed regarding the sharks behavior and why are needed. For example looking at figure one it appears that what changes is the area used by the animals, rather than simple residency. The el nino years show a larger KUD. So do the sharks use a larger area during 'stressful' times or do they actually alter their residency to the atoll? This distinction needs to be clarified.

This comment, and similar comments from R3, really helped to steer our revision of this work, thanks. We agree that more work was needed to try and explain how shark behaviour appeared to change under increased stress (rather than just the fact that it does). We realised that we were confusing the matter talking about residency and yet showing an indicator of space use (KUDs). We now include much more detail exploring both of these metrics, as well as exploration of absences from the array to show three much clearer results overall:

- 1) Grey reef sharks reduce their residency to particular parts of the reef (possibly their ‘central places’) in response to increase stress on the reef. This manifests itself as...
- 2) More diffuse space use across the reefs within our monitoring location AND
- 3) significantly longer periods away from our array (and by extension the shallow forereefs) when stress is high.

In summary, sharks appear to respond to high stress by moving more, occupying a larger areas and spending more time away from the reef. We hope you agree that these additional results improve the clarity and evidence base for our claims. These points are made clearly now throughout the manuscript.

I would also be a bit concerned that some of the results are related to correlations between years. For example, Jan-Jun 2014 was not an El Niño year, yet the UD sizes and residency analysis (Fig 1 and Fig 3) are almost identical to the El Niño year results. This may suggest that 2014-2016 results may have been driven by some other factor. Was 2013 an El Niño year?

We agree with the reviewers and a similar point also made by R1. To address this, we have included more data from non-El Niño years in the analysis, and have run the analysis on a dataset the El Niño years omitted. We still find the same pattern, suggesting that the result isn't driven by extreme values and correlations between years, but is driven by the changing environmental stress in the region. The additional analyses/results are detailed on lines 116 and 482.

Line 116 - A similar relationship between residency and combined environmental SE index (estimate = -0.07, $z = -4.34$, $p < 0.001$) was found even after data from El Niño periods were removed, suggesting persistence of this trend even without extreme climatic events known to cause high environmental stress to coral reefs.

Line 482 – To test whether the relationship between stress and residency were not exclusively driven by extreme stress events, a secondary analysis removing data from El Niño periods was undertaken.

Is Fig.4 needed? I am not sure what additional information is gained
Figure 4 has now been moved to the supplementary materials.

Discussion:

Line 203: I would say there is a lot of individual variability with depth use but some grey reef sharks may spend considerable time below 50 m (see Papastamatiou et al. 2018).

We thank the reviewer for this point, and this perhaps helps explain our new analysis looking at the absences from our array. This sentence has been updated to reflect this, line 245.

Line 245 - Reef sharks are ectotherms and have been seen to exhibit behavioral thermoregulation to regulate their body temperatures and avoid physiological damage from adverse SSTs¹⁵.

The final big issue is it would be nice to try and see what specifically from the 'stress' measure is causing changes in shark behavior. While I understand why there is the need for a single metric, it becomes less informative when the specific variables playing a role are unknown. As an example, cloud cover (one of the variables included) is not likely to cause changes in shark residency. While they do say they cant tease those apart, do they have access to he raw data that went into the single metric? Or is the index the only processed value they received.

This point aligns with comments from R1. We have updated the manuscript to provide additional information on why we believe it was preferably to use an index that we developed in Williamson et al. 2022 to evaluate this relationship, rather than single metrics, primarily due to the fact that environmental stress can be driven by different environmental factors even within an atoll (see responses to R1). As such, to assess the impact of how environmental stress on coral reefs can impact shark residency and space use, we feel it's more appropriate to use an index that includes multiple variables, rather than singular variables, the impact of which may vary from location to location at this site. We do however, discuss the individual variables potentially driving this relationship more in the discussion (lines 218, 242).

Lines 218 - The spatial variation in residency observed could also be driven by hydrodynamic factors. Coral response to environmental stress, such as bleaching, can be highly variable, even within a reef system, and often is the result of differing fine scale environmental and biological processes^{32,43}. There is also some congruence between areas of increased residency and areas that are sheltered from wave exposure. Shelter from wave exposure is associated with increased coral cover and quicker recovery from bleaching events⁴⁴, another potentially important factor influencing shark behavior.

The mechanisms driving these results are clearly complex and involve a mixture of variability in shark behavior as well as heterogeneity in coral reef response to environmental stress, and at different temporal scales.

Line 242 - This study did not examine the precise environmental factors driving reductions in residency. Stress on coral reefs is often closely linked to SST, and other temperature metrics, such as Degree Heating Weeks (DHW) and SST variability^{32,52}, and metrics of SST contributed considerably to the environmental SE index³². As such, the reduced residency found in this study could be driven by increases or changes in metrics of SST. Reef sharks are ectotherms and have been seen to exhibit behavioral thermoregulation to regulate their body temperatures and avoid physiological damage from adverse SSTs¹⁵. Therefore, an influence of different metrics of SST on movement is to be expected. Indeed, links with SST, SST anomalies, and SST variability and movement, residency, and presence/absence of other shark species have been seen elsewhere^{53,54}. For example Ryan et al.⁵⁴ found that low SST anomalies increased white shark (*Carcharodon carcharias*) presence and residency, which increased the chances of attacks on the eastern Australian coast. However, little is known about these relationships in reef sharks⁶. Season was also found to have a significant effect on residency in grey reef sharks, which supports previous research at this site that showed that grey reef sharks spent more time away from reefs during the wet season compared to the dry season²⁸. These changes in residency with season could be due to environmental or ecological factors. Shark species have been seen to increase movement and decrease residency during storm events^{55,56}, which may be increased during the wet season. Alternatively, residency changes may be due to changes in food resources, with historical fisheries known to peak in the wet season in this region^{57,58}. In addition, our results confirm that year is a variable that should be regularly included as a predictor variable to account for temporal variation when modelling movement ecology of marine species^{59,60}, which here is most likely linked particularly to the severity of environmental change associated with El Niño events.

Reviewer #3 (Remarks to the Author):

This manuscript uses acoustic telemetry data from grey reef sharks tagged in the Chagos Archipelago and satellite derived environmental data that is indicative of coral reef stress to test the relationship between the residency of sharks and the stress experienced by coral reefs. Given the increasing stress that climate change is placing on coral reefs, understanding such effects will be of importance to future management of sharks in reef environments. I consider the premise of the manuscript to be a good one – this is a sensible concept to investigate. However, I think that there is a little more work to do to get the manuscript to a point where it makes a compelling case for the relationship that was identified. My major points are:

A. The SE Index (environmental stress indicator) is a set of environmental parameters that appear to be used to show the level of stress that a coral reef is facing. As used in the current manuscript the index is calculated at the smallest possible spatial scale (50 m radius) to correspond to the location of acoustic receivers (more on the spatial scale of this coverage later). Where I am struggling is that nowhere that I could find in the manuscript was there any calibration as to what a particular level of the index means. For example, is there a value at which bleaching would occur? Ultimately, the reader has no way of knowing what a value of 0.2, or 0.4, of the index means to a reef. One is obviously worse than the other, but is there a linear relationship in this relationship? The addition of this type of information is critical to the reader understanding what the index means and so they would be able to get a sense of what this might mean for the sharks. From fig 1 it appears that the mean(?) Values to do exceed 0.4, and are often around 0.2. what I really wanted to know was how bad was 0.4 for reefs?

We thank the reviewer for their point, and we agree that we can get this across better. From previous research we have seen that greater RESET values do occur during bleaching events (Figure RR3,

from Williamson et al. 2022), indicating that environmental stress is greater on reefs during these periods (we believe our new Fig. 1 helps to convey this point as well). Coral degradation is ultimately caused by a complex of biological and environmental factors, in which there are currently significant knowledge gaps. And there is both considerable intra-, and inter-, reef variation in the drivers of incidents such as bleaching. It is very unlikely you could equate a single stress score to a reef response such as bleaching. As such, our index cannot be used to directly quantify the health of coral reefs, as discussed in Williamson et al. 2022. Rather, it is a temporally explicit monitoring tool (i.e., to compare against various time periods from the same region) to evaluate relative changes in stress exposure on coral reef ecosystems.

Figure RR1 Mean monthly combined environmental SE scores taken quarterly from all sites (January, April, July and October) from the central Saudi Arabian Red Sea, the Chagos Archipelago and the Gilbert Islands in the Republic of Kiribati; and from 01/01/2003 to 31/12/2016. Trend line in blue with 95% standard error in grey. El Niño events are highlighted by strength; strong = red; moderate = orange; and yellow = weak. Asterisks note El Niño years where bleaching events occurred during this period (reproduced from Williamson et al. 2022).

However, from research at this site, we know that El Niño periods, which caused significant bleaching, regularly had values over 0.4, so we can say, at this site, and during these periods, that RESET values 0.3/0.4+ indicate considerable stress to the reef. In the Chagos Archipelago, these values have resulted in localised bleaching events which have been documented fairly extensively elsewhere in the literature. We have added information to the manuscript (line 63, line 415) that reflects the above points.

Line 63 - Environmental stress can be defined as negative impacts on the growth and health of ecosystems resulting from changes or extremes in environmental variables^{29,30}. Coral reefs are susceptible to a number of environmental stressors³¹, which in turn may impact reef shark populations. However, there can be significant inter-, and intra-regional variance in how different

environmental variables drive stress on these ecosystems^{32,33}. For example, an environmental stress index, based on satellite remote sensing data that allows assessment of multiple abiotic environmental stressors, recently found that sea surface temperature (SST), current and wind were the primary drivers of environmental stress in the Chagos Archipelago in the Indian Ocean, however depth and SST, and Degree Heating Weeks, SST, and current, were stronger drivers of stress on coral reefs in the Red Sea and the Gilbert Islands, respectively³². Composite indices such as this, therefore, capture interactive variables that may increase, or reduce, environmental stress, providing an opportunity to gain a more holistic understanding of how multiple environmental stressors on coral reefs can impact reef shark movement and residency.

Line 415 - From previous research at this site, RESET scores of 0.3 or higher indicate considerable stress to the reefs in Chagos³².

B. As mentioned above, the pixel size for the index was 50 m to represent the location of each of the receivers. However, the manuscript talks about the likely acoustic detection range being 300-500 m (radius). I wonder whether calculating the index on the area that corresponds to the detection range would be more appropriate since this is the area that sharks detected would occur in. I suspect this will make little difference since the index is likely to vary little over such small spatial scales. However, I could be wrong, and I think it would be useful to explore using this slightly broader area for the index.

This is a valid point and we have updated the data set, and increased the pixel size for the RESET to 500m as suggested. The methods have been updated to reflect this, see line 398.

Line 398 - As the spatial resolution of the nine variables varied (Supplementary Table 5), each product was resampled using bilinear interpolation to match the detection range of the receivers (500m)⁸⁷

C. I'm struggling with the kernels shown in fig 1. I assume that these are the kernel distributions of the shark locations of those detected in the relevant time period. This is a pretty standard approach. However, the figure caption says that these show "shark residency behavior". However, I don't see the direct connection here. Assuming these are occurrence data it shows where the animals were mostly detected, not what their residency was. Given the detection ranges of the receivers the kernels overestimate the space use of the sharks, especially when sample sizes are small (up until July-Dec 2015). By this I mean that the areas shown in the kernels include mostly area that is not covered by receiver detection ranges. The change in size of the kernels with sample size could be overcome by using a fixed smoothing factor (h) for the kernel estimation (ie the same for all periods rather than using an estimator that changes with each time period).

The issue that I see here is that some readers may have the impression that the sharks are more widely moving in the early period dominated by El Nino conditions because of the larger size of the kernels. Given the sample size and smoothing factor issues this is not a certainty. However, I think the bigger question is what it is the kernels are meant to be showing and whether they have a place in the manuscript. It remains unclear to me how they help the reader interpret the residency index values. Thus at the very least there needs to be more information provided on what they show, and what it means for the results on residency that they are meant to demonstrate.

We thank the reviewer for this helpful comment. We have now thoroughly revised our approach and consider area use and residency separately now (see responses to R2 comments above that outlines our three key results that help to clarify our manuscript). In terms of sample size we have now added more data at either end of the time-series. There are still six-month periods that have fewer individuals at liberty but this increased data set, we believe, now gives us more rigour. We have also replaced Fig. 1 (see above) which we feel is now much more informative and better represents our key results.

D. Related to the sample size issue identified above (ie smaller sample sizes up to Jul-Dec 2015) is what is displayed in fig 3. There is a clear change in distribution and size of the points when samples

sizes increase. Given this I think it is hard to infer too much from this figure and I suggest it would be better to plot this data in another way that is not affected by sample size, and that would allow for some statistical testing of the distribution of the values between periods. It is possible that there is a distinct change in residency index distribution during this period, but without removing the sample size effects it is hard to conclude such a result.

We have now added more data, as discussed, which should help partially address this issue. The rationale for using proportions in the first place was to avoid too much bias from the sample size issue in early years, which unfortunately we can't do much about. That said we now have the additional analyses that compliments this representation of individual variation – as discussed, we show a statistically significant increase in time spent away from the reef during times of high stress. The median duration of time spent away from the forereefs were not stochastically equal between times of low and high stress; grey reefs were absent for significantly longer when stress was high (Brunner-Munzel; $\hat{P}^*(1235.9) = -2.8336$, $p = 0.0047$). The probability that sharks would remain away from the forereef longer during times of stress was 0.4661 (see new Fig 3 below.).

Figure 3. Variation in absence and individual shark residency across an eight-year period. A The mean delay in log days, between detections for individuals leaving the forereef (note: for clarity we represent the mean but test the median using a Brunner-Munzel test to show that the probability that sharks would remain away from the forereef longer during times of stress was 0.466). B The proportion of tagged individuals falling within each mean residency index bin (0.0 - 1.0) across 14 sixth monthly periods. Yellow/red indicate El Niño conditions and blue, non-El Niño conditions.

E. One of the important considerations in this work is where do the sharks go when they are not detected by the receivers? This is explored a little in the discussion, but I think is a more central part

of the consideration of what this all means. And it may also be an area that further analysis might be able to help reveal. The manuscript implies that sharks travel considerable distances when environmental stress increases. This is not an unreasonable conclusion, but not one that is well explained or well tested in the manuscript. The difficulty is that the acoustic receivers only provide particle presence data and there is no information on where they have gone. So it is equally plausible that when a shark is not detected that it is just outside the detection range of the receiver (e.g. moved down the reef into deeper water where the receiver cannot hear the signal), within the detection range but not detected for some reason (e.g. signal interference, environmental conditions) or a long way from the receiver. Thus the conclusion that sharks are traveling considerable distances is one of a number of equally plausible conclusions. I see a couple of options for how to dig more deeply into this issue. Firstly, the residency index is calculated individual receivers, and measures the proportion of days each month that an individual is present. So it would also be possible to calculate the mean/max/etc number of continuous days absent by month. If there is an increase in the value of this index, then I would interpret this as the likelihood that sharks have moved farther away from the receiver. However, if this value does not change, then it suggests that sharks are moving away for similar periods and possibly similar distances as when they are more resident. The other potential test would be to determine the likelihood that individuals moved to other receivers during periods of lower residency. If the animals are detected at other receivers and farther away during periods of lower residency this would be good evidence of movement of considerable distances. By considering both of these analyses in combination, you could get an indication of where they may have moved (a) if they were detected at other receivers this might suggest that they are moving to other reef locations (possibly with lower env. stress), (b) if they did not get detected at other receivers, but did have longer periods of continuous absence this might indicate movement to areas farther from the reef or into deeper water. The depth question might be answerable if the data had depth sensors, but the methods do not mention them so I assume they did not and so this could not be investigated. This type of exploration would provide a lot more depth to this manuscript and help turn the Discussion from what is largely speculative about a range of issues that are currently untested.

This is a really nice suggestion and we thank the reviewer for this insightful input. As mentioned in the previous response, we have now conducted some additional analyses to this effect, demonstrating that during periods of high stress (El Niño periods), the median time spent away from the reef, between visits to any one of our receivers, is significantly higher than when stress is low (see above). While we also suspect that this is due to sharks spending more time at depth, we have been more careful in our interpretation in the discussion, only focusing on what we can actually say for sure and ensuring any speculation is clearly indicated as such.

F. Related to the previous point is the possibility that the residency of sharks did not change over time, but their detectability did (and hence the number of days detected per month). The variation in the detection ability of acoustic receivers with a wide range of environmental conditions has been tested a reasonable amount (e.g. Huvneers et al., 2016) and shown to vary quite a lot in certain situations. As such this is a factor that should be addressed in the current manuscript. For example, there was a season difference in residency according to the model. I assume one of these was a windy monsoon season. Wind driven wave action can be a factor that effects detectability, and so may have been a factor. Of course it could also be the opposite, so context here is important. Testing these issues can be quite tricky, and so I am not suggesting to do much beyond acknowledging that this may be a factor that needs to be considered in the interpretation of the data. The authors may have some way to test this as well, which would be even better.

The reviewer is correct, our results could be impacted by detectability, and we have added some details to the discussion to reflect this point, line 193.

Line 193 - From a practical perspective, changes in residency may also be due to the influence of changing environmental conditions, such as wind speed, on acoustic detectability^{37,38} in addition to coral reef health. Range testing was not feasible at this site during the period of study, so this could not be assessed. However, given the long time-series of data obtained, and the wide variation in environmental conditions throughout the study period, the impacts of varying detectability is like to be minimal.

G. As mentioned above, the Discussion is largely speculative because of the types of data and results generated. Hence the suggestion that some additional exploration of the data be undertaken to help provide some way of reducing this speculation. The temptation is always to assume that there is an important and meaningful reason for the patterns found, but it is important to try and provide some additional information that may support these over other types of conclusions. I'm not suggesting that all speculation is removed from the Discussion. Some of this is useful and warranted. However, it needs to be presented in a way that the reader understands that there is a range of plausible outcomes. So statements such as "These findings will likely have important repercussions for trophic interactions and reef ecosystem functioning" (line 128) should be tempered a bit and the full range of possibilities provided.

We have taken this point on board and have revised the discussion in a number of places to address this. With the additional analyses, we also feel that we have a slightly clearer understanding now of how behaviour might be altered in response to stress. Of course, there are a whole host of new questions raised around direct and indirect mechanisms but these are things we hope to chip away at over the next few years drawing in additional habitat data etc. For these reasons though, we have tempered the speculation in the discussion.

H. In the paragraph starting on line 196 the effect of environmental factors on the movement and residency of sharks is explored. This is a useful paragraph. However most of the examples come from non-reef species. The statement "little is known of these relationships in reef sharks" is made. While there is not extensive knowledge there are some studies and I think these should be considered here. Some of these conclude that environmental parameters have limited effects on reef sharks, which would seem like a useful piece of information for the reader to have. Examples that I am familiar with include: (Heupel & Simpfendorfer, 2015), (Schlaff et al., 2017), (Espinoza et al., 2015). On line 217 the observation is made that sharks increase movement during storm events. This is mostly true, but Udyawer et al. (2013) reported that a reef shark species (blacktip reef sharks) did not move during a major storm when all other species did.

This is a good point, and we have added this detail and additional references into this section of the discussion, line 253. We have also added further information on movements due to storms as referenced in line 267.

Line 253 - However, little is known about these relationships in reef sharks⁶, and the few studies that have investigated these relationships have typically found that changing environmental conditions have limited impact. For example, Schlaff et al.⁵⁵ found that size and sex were the most important drivers of activity space in Australian blacktip reef sharks *Carcharhinus melanopterus*, with salinity and water temperature having significant but relatively low impacts, while Heupel & Simpfendorfer⁵⁶ found no relationship between activity space and environmental variables in grey reef sharks on the Great Barrier Reef. As such, these results, to our knowledge, provide some of the first evidence of changing environmental variables impacting the movement and residency of grey reef sharks.

Line 267 - Shark species have been seen to increase movement and decrease residency during storm events^{57,58}, which may be increased during the wet season.

I believe that addressing the above points will significantly improve the manuscript and provide more certainty about what the results show and how important they are for coral reef sharks.

Espinoza, M., Heupel, M. R., Tobin, A. J., & Simpfendorfer, C. A. (2015). Residency patterns and movements of grey reef sharks (*Carcharhinus amblyrhynchos*) in semi-isolated coral reef habitats [Article]. *Marine Biology*, 162(2), 343-358.

Heupel, M. R., & Simpfendorfer, C. A. (2015). Long-term movement patterns of a coral reef predator.

Coral Reefs, 34, 679-691. <https://doi.org/10.1007/s00338-015-1272-4>

Huveneers, C., Simpfendorfer, C. A., Kim, S., Semmens, J. M., Hobday, A. J., Pederson, H., Stieglitz, T., Vallee, R., Webber, D., Heupel, M. R., Peddemors, V., & Harcourt, R. G. (2016). The influence of environmental parameters on the performance and detection range of acoustic receivers. *Methods in Ecology and Evolution*, 7(7), 825-835. <https://doi.org/10.1111/2041-210X.12520>

Schlaff, A. M., Heupel, M. R., Udyawer, V., & Simpfendorfer, C. A. (2017). Biological and environmental effects on activity space of a common reef shark on an inshore reef. *Marine Ecology Progress Series*, 571, 169-181. <http://www.int-res.com/abstracts/meps/v571/p169-181/>

Udyawer, V., Chin, A., Knip, D. M., Simpfendorfer, C. A., & Heupel, M. R. (2013). Variable response of coastal sharks to severe tropical storms: environmental cues and changes in space use [Article]. *Marine Ecology Progress Series*, 480, 171-183.

REVIEWERS' COMMENTS:

Reviewer #1 (Remarks to the Author):

I am satisfied with the authors' reanalysis of their expanded dataset and think they have done a commendable job revising their manuscript. I only have a few very minor comments.

Minor comments

Table 1: Is 97.5% CI a typo for 95% CI (i.e., 2.5% and 97.5% bounds)?

Lines 481-483: Please include the range of dates that were excluded from analysis.

Reviewer #2 (Remarks to the Author):

The authors have made some major edits to their manuscript and I appreciate their detailed response to my comments and those of the other reviewers. The manuscript is greatly improved. In particular, figure 1 now clearly shows the change in movements associated with the major el nino event.

I have two comments:

- 1) The Figure 1 caption is not correct. The residency is not plotted against changes in UD's, the x-axis is month. So this figure (nicely) shows temporal changes in residency and UD's. Also, the figure caption doesn't say that residency is the blue line (I assume it is anyway).
- 2) Looking at figure 1 it's hard to see that residency and environmental stress index are negatively correlated. The strong el nino event instead seems to show a clear delay in increased residency by the sharks, while causing an immediate increase in space use area. Am I reading the figure incorrectly?

Reviewer #3 (Remarks to the Author):

The substantial revisions made to the manuscript have addressed all of my concerns. I congratulate the authors on a job well done.

REVIEWERS' COMMENTS:

Reviewer #1 (Remarks to the Author):

I am satisfied with the authors' reanalysis of their expanded dataset and think they have done a commendable job revising their manuscript. I only have a few very minor comments.

We thank the referee for taking the time to review our manuscript again and we're pleased that they are happy with our revisions. We have addressed their minor comments below.

Minor comments

Table 1: Is 97.5% CI a typo for 95% CI (i.e., 2.5% and 97.5% bounds)?

This is correct and thank you for picking this up. This has been corrected in the manuscript.

Line 796 - Table 1. GLMM results following model selection and model averaging for residency in grey reef sharks. Conditional results are presented. Estimates with unconditional standard error, 95% confidence intervals (CI), associated p values are presented, with significant results highlighted in bold.

Lines 481-483: Please include the range of dates that were excluded from analysis.

These dates have been added as requested.

Line 486 - To test whether the relationship between stress and residency were not exclusively driven by extreme stress events, a secondary analysis removing data from El Niño periods (01/07/2014 - 30/06/2016 and 01/07/2018 – 30/06/2019) was undertaken.

Reviewer #2 (Remarks to the Author):

The authors have made some major edits to their manuscript and I appreciate their detailed response to my comments and those of the other reviewers. The manuscript is greatly improved. In particular, figure 1 now clearly shows the change in movements associated with the major el nino event.

We thank reviewer again for their insightful comments and for being willing to review this revised version. We have now addressed your two remaining comments.

I have two comments:

1) The Figure 1 caption is not correct. The residency is not plotted against changes in UDs, the x-axis is month. So this figure (nicely) shows temporal changes in residency and UDs. Also, the figure caption doesnt say that residency is the blue line (I assume it is anyway).

That is correct and we thank the reviewer for picking this up. We have altered the to reflect the above points.

Line 801 – Figure 1. Impact of environmental stress on shark space use and residency. Temporal trends in combined environmental stress exposure (SE) index experienced by coral reefs in the northern atolls of the Chagos Archipelago during a strong El Niño ‘episode’ and weaker El Niño ‘conditions’ (A). Grey reef shark residency (blue trend line) during the same period (Feb 2013 – Feb 2021) and temporal changes in area use (km²) measured as the 50% (yellow), 75% (green) and 95% (pink) kernel utilization estimation (KUD)(B).

2) Looking at figure 1 its hard to see that residency and environmental stress index are negatively correlated. The strong el nino even instead seems to show a clear delay in increased residency by the sharks, while causing an immediate increase in space use area. Am I reading the figure incorrectly?

Thanks for this feedback. Your interpretation of the plot is correct. The immediate effects are increases in space use and residency has a delayed increase as stress starts to peak and come down again. Accounting for all the variability and the noise in the data, our model does demonstrate on average a significant negative relationship between residency and stress over the duration of the study, however, we acknowledge it is complex. From Fig. 1 the trend is perhaps more obvious when the changes in stress are not so extreme (i.e. during the 2018 El Niño conditions).

In response to this comment we have now altered the sentence in the results as follows:

Line 106 – A significant negative relationship between residency and combined environmental SE index was found (estimate = -0.1, z = -10.48, p <0.001, Fig. 1), indicating that on average across the reefs of the northern atolls of the archipelago, grey reef sharks became less resident as environmental stress on reefs increased, particularly during strong El Niño conditions, albeit with a delay in these effects during the strong El Niño event. Kernel estimates (KUD) of core (50%), 75% and 95% space use all increased almost immediately during elevated periods of stress exposure, suggesting space use became more diffuse (Fig. 1B).

We hope that this addresses this comment sufficiently.

Reviewer #3 (Remarks to the Author):

The substantial revisions made to the manuscript have addressed all of my concerns. I congratulate the authors on a job well done.

We thank the reviewer for this and their previous assessment – the feedback from all referees has been extremely helpful in strengthening our paper.